# Origins of Barents-Kara sea-ice interannual variability modulated by the Atlantic pathway of El Niño–Southern Oscillation

Binhe Luo[1], Dehai Luo[2,3] ✉, Yao Ge[2,3], Aiguo Dai [4], Lin Wang [5], Ian Simmonds [6], Cunde Xiao[1] ✉, Lixin Wu[7] & Yao Yao [2,3]

Winter Arctic sea-ice concentration (SIC) decline plays an important role in Arctic amplification which, in turn, influences Arctic ecosystems, midlatitude weather and climate. SIC over the Barents-Kara Seas (BKS) shows large interannual variations, whose origin is still unclear. Here we find that interannual variations in winter BKS SIC have significantly strengthened in recent decades likely due to increased amplitudes of the El Niño-Southern Oscillation (ENSO) in a warming climate. La Niña leads to enhanced Atlantic Hadley cell and a positive phase North Atlantic Oscillation-like anomaly pattern, together with concurring Ural blocking, that transports Atlantic ocean heat and atmospheric moisture toward the BKS and promotes sea-ice melting via intensified surface warming. The reverse is seen during El Niño which leads to weakened Atlantic poleward transport and an increase in the BKS SIC. Thus, interannual variability of the BKS SIC partly originates from ENSO via the Atlantic pathway.

Over the Arctic, the sea ice concentration (SIC) exhibits a decline trend throughout the year[1–4] and notable variability on interdecadal[5,6] and interannual[7,8] timescales. Arctic SIC loss has been recognized as a major driver of the Arctic amplification in autumn and winter[9,10]. Arctic amplification not only influences Arctic ecosystems[11], but may also affect winter atmospheric circulation and associated cold extremes over the Northern Hemisphere continents[12–14], although Arctic amplification's impact is confined mainly over the northern high latitudes[15]. The recent decreasing trend and interdecadal variations of Arctic SIC have been shown to be related to the increasing $CO_2$[16] and Atlantic multidecadal Oscillation[5]. Interannual variations of the winter Arctic SIC are stronger in the Barents-Kara Seas (BKS) than in other Arctic regions[17]. However, what causes the strong winter SIC's interannual variations over the BKS is unclear. Further, it is unknown whether there are any changes in winter SIC's interannual variability over the BKS in

the recent decades. The El Niño-Southern Oscillation (ENSO) is a primary source of interannual variability[18], yet little is known on the relation between the interannual variability of the winter BKS SIC and ENSO.

Variations in winter Arctic SIC have been considered as a response to atmospheric and oceanic forcings[19–23], although the SIC's variations provide a strong positive feedback through changes in surface fluxes that amplifies the response to the forcing[24]. On interannual and longer timescales, BKS SIC changes are partly due to the Atlantic ocean heat inflows into the western Barents Sea as seen from observations and model simulations[20–22]. The intrusion of atmospheric warm and moist air into the BKS associated with the combination of Ural blocking events with the positive phase (NAO⁺) of the sub-seasonal (10–20 days) North Atlantic Oscillation (NAO) events can also melt winter sea ice there[19] mainly through increased downward infrared radiation and

[1]State Key Laboratory of Earth Surface Processes and Resource Ecology, Beijing Normal University, Beijing 100032, China. [2]Key Laboratory of Regional Climate-Environment for Temperate East Asia, Institute of Atmospheric Physics, Chinese Academy of Sciences, Beijing 100029, China. [3]University of Chinese Academy of Sciences, Beijing 101499, China. [4]Department of Atmospheric and Environmental Sciences, University at Albany, State University of New York, Albany, NY 12222, USA. [5]Center for Monsoon System Research, Institute of Atmospheric Physics, Chinese Academy of Sciences, Beijing 100029, China. [6]School of Geography, Earth and Atmospheric Sciences, University of Melbourne, Parkville, VIC 3010, Australia. [7]Frontiers Science Center for Deep Ocean Multispheres and Earth System and Key Laboratory of Physical Oceanography, Ocean University of China, Qingdao 266100, China and Laoshan Laboratory, Qingdao 266237, China. ✉e-mail: ldh@mail.iap.ac.cn; cdxiao@bnu.edu.cn

enhanced surface warming[25–27]. The transport of Atlantic ocean heat toward the high latitudes is linked to North Atlantic sea surface temperature (SST) anomalies associated with the Atlantic meridional overturning circulation (AMOC)[28]. The AMOC shows notable inter-decadal and interannual variability[29–31]. While the interdecadal fluctuations of the AMOC can be characterized by the Atlantic Multidecadal Oscillation (AMO)[32], the interannual AMOC is an interannual North Atlantic SST tripole mode coupled with the interannual NAO-like pattern via the adjustment of oceanic internal Rossby waves or deep convection to surface wind stress or buoyancy forcing[33,34].

Although the AMO and tropical North Atlantic mean SST can modulate ENSO in the Pacific[35,36], numerical model experiments indicate that ENSO may remotely influence interannual variability of North Atlantic SST associated with AMOC via NAO-like patterns in surface winds and turbulent heat fluxes[37]. Because North Atlantic SST anomalies can influence the BKS SIC via both the Atlantic ocean heat transport[20–22] and atmospheric moisture intrusion[19,26,27], we hypothesize that interannual variations in the BKS SIC may partly originate from the remote influence of ENSO. Here, we use a 9-year high-pass filter to remove the decadal and longer variations in winter (December-January-February, DJF) SIC, SST and Niño3.4 index to retain their interannual (≤8 years) variations. Then, the leading empirical orthogonal function (EOF1) of the high-pass filtered DJF SST anomalies over the North Atlantic (80°W–0°, 20°–70°N) and its corresponding principal component (PC1) are used to quantify the interannual variability of the North Atlantic SST tripole mode (Methods), which is referred to as the Atlantic Interannual Variability (AIV). We show that ENSO can influence BKS SIC via its impact on the Walker circulation from tropical Pacific to Atlantic, which in turn influences Atlantic Hadley cell, leading to an NAO-like anomaly circulation pattern that affects poleward heat transport into the BKS by the Atlantic ocean and atmosphere. Further,

we demonstrate that recent increases in ENSO amplitude have contributed to the enhanced interannual variability in recent winter BKS SIC.

## Results

### Increased interannual variability of BKS sea ice in recent decades and its link to ENSO

We divide our analysis period 1950–2019 into two sub-periods: 1950–1989 and 1990–2019 to compare the interannual variability of the winter BKS SIC, as Arctic warming is accelerated after 1990[9,10,12]. DJF SIC anomalies averaged over the BKS (30°–90°E, 65°–85°N) (Fig. 1a) clearly show notable interannual variations with 4–8 years. This is easily seen from the power spectrum of its normalized time series during 1950–2019 for the detrended (Fig. 1b) and 9-year high-pass filtered (Fig. 1c) data. The standard deviation of the filtered BKS SIC time series is 0.58 during 1950–1989 but increases to 1.51 during 1990–2019 (Fig. 1a). Excluding 1950–1964, which had little variation possibly due to data issues, increases the standard deviation to 0.75 during 1965–1989, which corresponds to a 101% increase from 1965–1989 to 1990–2019. Thus, the amplitude of the interannual variability in winter BKS SIC has more than doubled from 1950–1989 to 1990–2019, which is also true if the 1950–1979 and 1980–2019 periods are compared. The large increase in the SIC variability is also seen in ERA5 reanalysis data (Supplementary Fig. 1) and CMIP6 data (Supplementary Fig. 2), but has not been reported in previous studies[7,8].

We find that the standard deviation of the 9-year high-pass filtered DJF Niño3.4 SST index increases from 0.91 during 1950–1989 to 1.09 during 1990–2019 (Fig. 2a), thus representing a ~20% intensification of the ENSO variability. Such an increased ENSO variability is characterized by more frequent extreme (or large-amplitude) El Niño and La Niña events in the recent decades (Fig. 2a), and is consistent with

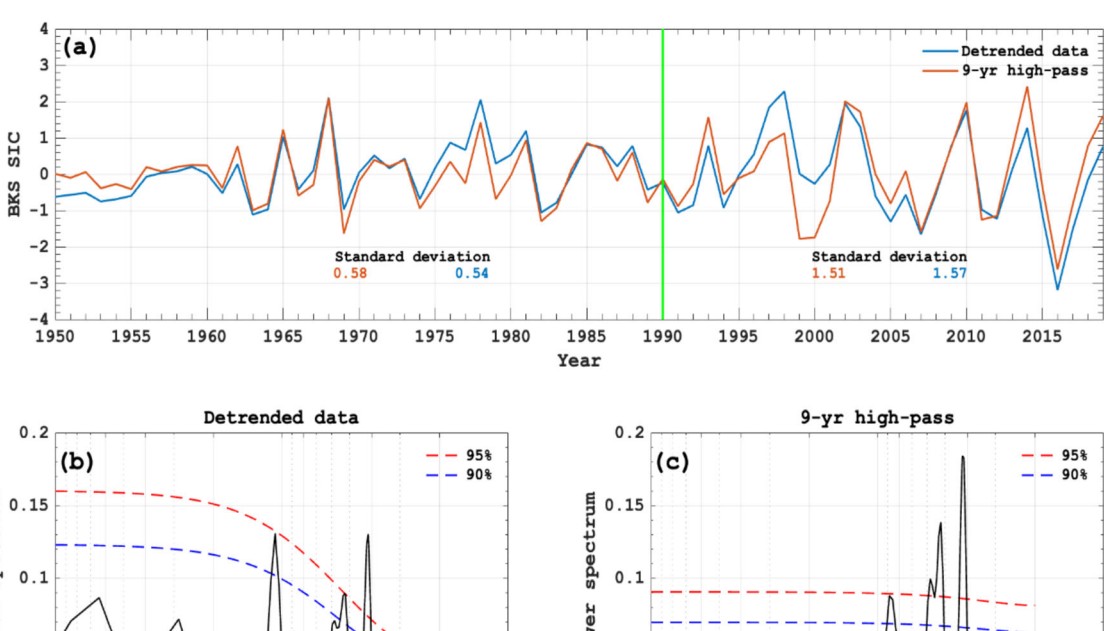

**Fig. 1 | Temporal variability and power spectra of the winter Barents-Kara sea-ice for detrended and 9-year high pass filtered data. a** Normalized time series of DJF (December-February) mean sea ice concentration (SIC) anomaly (%) averaged over the Barents-Kara Seas (BKS) (30°–90°E, 65°–85°N) for linearly detrended (blue line, no filtering) and 9-year (or 9-yr) high-pass filtered (red line) SIC data from the

Hadley center, where the value in the left- (right-) hand side of the green vertical line represents the standard deviation of the DJF-mean BKS SIC variation averaged over 1950–1989 (1990–2019). **b, c** Power spectra of the **b** detrended and **c** 9-yr high-pass filtered DJF-mean BKS SIC time series shown in (**a**). In panels **b–c**, the blue (red) dashed line represents the 90% (95%) confidence level.

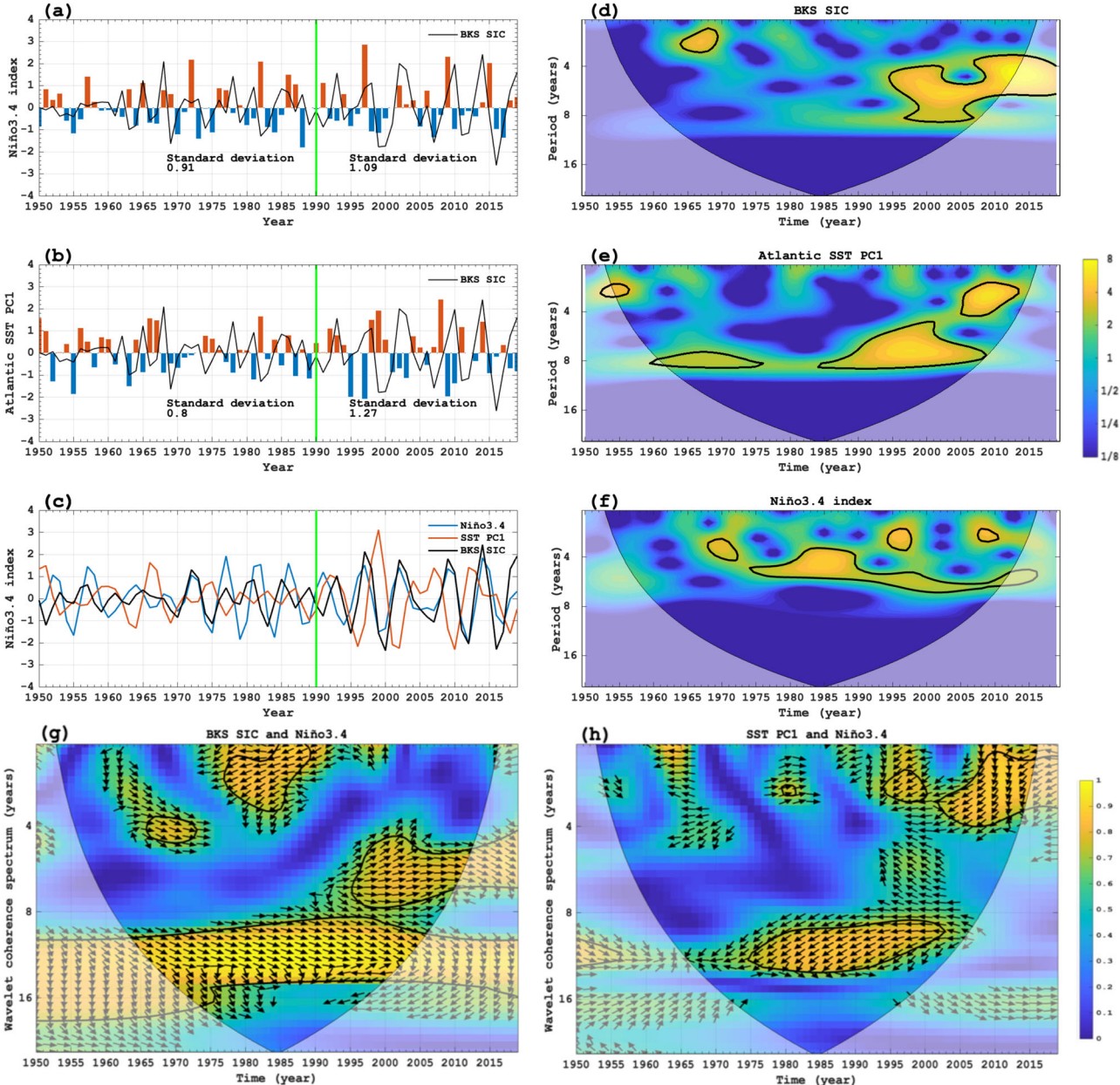

**Fig. 2 | Wavelet power spectra and coherence spectra of the temporal variations of the winter sea ice concentration (SIC) over Barents-Kara Seas (BKS), North Atlantic sea surface temperature (SST) and Niño3.4 index.**
**a**, **b** Normalized time series of **a** 9-year high-pass filtered DJF (December-February) mean Niño3.4 index (bars) and **b** the first principal component (PC1, bars) of the leading Empirical Orthogonal Function (EOF1) mode of the 9-year high-pass filtered DJF-mean SST anomalies over the North Atlantic (80°W–0°, 20°N–70°N) during 1950–2019. The black solid line in **a**, **b** is the normalized time series of DJF-mean SIC (%) anomalies averaged over BKS (30°–90°E, 65°–85°N). **c** 4–8-yr band-pass curves of the normalized DJF-mean BKS SIC (black), Niño3.4 (blue) and SST PC1 (red) time

series. **d**–**f** Wavelet power spectra for 9-year high-pass filtered normalized DJF-mean **d** BKS SIC, **e** Atlantic SST PC1 and **f** Niño3.4 index time series during 1950–2019, where the ordinate is the Fourier period (year) and the abscissa is time. The thick contour encloses regions with at least 95% confidence and the thin line indicates the limit of the cone of influence. **g**, **h** Wavelet coherence spectra of 9-year high-pass filtered normalized **g** BKS SIC and **h** SST PC1 time series with the Niño3.4 index. The arrows in the significant regions indicate the phase relationship between the BKS SIC or SST PC1 and Niño3.4 time series with the in-phase pointing right (antiphase pointing left) and the Niño3.4 leads the BKS SIC (SST PC1) in the 4–8-year (2–5-year) bands during 1996–2013 (1994–2013).

increased ENSO amplitudes under increased greenhouse gases[38–42]. The PC1 time series of the extratropical North Atlantic 9-year high-pass filtered SST, called Atlantic Interannual Variability (AIV) here, has a standard deviation of 0.80 during 1950–1989, which increases to 1.27 during 1990–2019 (Fig. 2b), indicating a ~59% increase. This enhanced AIV during 1990–2019 is related to the increased variability of the recent ENSO amplitude, as the correlation coefficient with the 9-year high-pass filtered Niño3.4 index changes from −0.049 ($p > 0.1$) during 1950–1989 to −0.51 ($p < 0.01$) during 1990–2019. Thus, the ENSO-AIV connection is significantly intensified during 1990–2019.

Previous studies[43,44] indicate that ENSO can influence the North Atlantic via changes in the Walker circulation between the tropical Pacific and Atlantic basins and the Hadley circulation. In particular, strong ENSO events can enhance AIV associated with interannual AMOC variability via the intensified NAO-like pattern[37]. While the NAO-like pattern has a significant negative correlation with the ENSO in the 4–8-year band[45], the atmospheric response to the ENSO variability is nonlinear[46–48]. As a result, the interannual SST anomalies associated with AMOC in response to surface wind stress forcing and turbulent heat fluxes[37] of the winter NAO-like pattern

exhibit a complex nonlinear behavior[30,46–49] different from that of the ENSO.

We also see that for 4–8-year band-pass filtered data the variability of the BKS SIC, AIV and ENSO is more notable during 1990–2019 than during 1950–1989 (Fig. 2c). While the BKS SIC has a modest positive correlation of 0.18 ($p > 0.1$) with the ENSO during 1950–2019, their correlation is non-stationary, changing from −0.01 during 1950–1989 to 0.32 ($p < 0.05$) during 1990–2019. This correlation increases to 0.2 (0.63) during 1950–1989 (1990–2019) for 4–8-year band-pass filtered data. Thus, the ENSO-BKS SIC connection is significantly strengthened during 1990–2019. This strengthening results from enhanced Atlantic ocean heat transport[50–53] and atmospheric intrusion of warm and moist air from the Norwegian Sea into the BKS[19]. Although the AIV has a negative correlation of −0.25 ($p < 0.1$) with the BKS SIC during 1950–2019, their correlation is non-stationary and −0.13 (−0.29, $p < 0.1$) during 1950–1989 (1990–2019) and becomes −0.32 (−0.39) for 4–8 years band-pass filtered data. Thus, the AIV-BKS SIC linkage is also strengthened in recent decades.

The wavelet power spectra of the BKS SIC, Niño3.4, and Atlantic SST PC1 indices (Fig. 2d–f) show the time lag of the strong SST PC1 (BKS SIC) variability behind the Niño3.4 index variability. The BKS SIC has significant power in the 4–8-yr band during 1995–2019 (Fig. 2d), whereas the SST PC1 shows significant power in the 4–8- and 2–4-yr bands mainly during 1990–2013 (Fig. 2e). The strong ENSO variability appeared in the 4–8-yr and 2–4-yr bands mainly during 1980–2015 (Fig. 2f). These results suggest that an enhanced ENSO variability starts ~10 years earlier than the strengthened AIV, which is ~5 years earlier than the enhanced BKS SIC variability (Fig. 2d). The causal relationship between the ENSO and the AIV or BKS SIC can also be revealed by calculating their wavelet coherence spectra. In the wavelet coherence spectrum map of two time series for the first and second variables, arrows pointing to the right-down or left-up indicate that the first variable is leading, while arrows pointing to the right-up or left-down show that the second variable is leading[54–56]. Thus, Fig. 2g–h further reveal that the ENSO leads the BKS SIC by ~15 years mainly in the 4–8-yr band during 1996–2013, whereas it leads the AIV mainly in the 2–5-yr band during 1994–2013. The main cause of this lag is probably that the AIV is a delayed response to ENSO-induced wind stress or buoyancy forcing of the winter NAO-like pattern in the North Atlantic mid-latitudes, where strong SST anomalies lag the wind stress or buoyancy forcing by several years[34,49]. The Atlantic ocean heat transport associated with the AIV also needs several years to reach the BKS[52,53] so that the ocean-induced BKS SIC variability lags the AIV.

Regressed fields (Eq. 1 in Methods) show that during 1990–2019 the BKS SIC decreases during a La Niña-like SST anomaly over Pacific basin and a cold-warm-cold SST tripole pattern emerges in the North Atlantic mid-high latitudes (Fig. 3d). This SST pattern resembles the regressed SST anomaly fields associated with the La Niña (Fig. 3e) and the positive phase of AIV (Fig. 3f). During 1950–1989 the SIC decrease needs a cold-warm-cold SST tripole pattern, which is not associated with a typical La Niña (Fig. 3a). Thus, the ENSO-BKS SIC connection is weak during 1950–1989 due to a weak ENSO-AIV linkage (Fig. 3b, c), even though the BKS SIC loss also requires a positive phase of AIV during 1950–1989 (Fig. 3i). The regressed winter Z500 and SAT anomalies onto the BKS SIC, Niño3.4 and SST PC1 time series further show that the La Niña corresponds to a positive-phase winter NAO (NAO⁺)-like pattern during 1950–2019 (Supplementary Fig. 3) and during its two subperiods: 1950–1989 and 1990–2019 (Fig. 3h, k). However, the NAO⁺-like pattern is stronger during 1990–2019 (Fig. 3k) than during 1950–1989 (Fig. 3h). Thus, the connection of the NAO-like pattern to ENSO is weak (strong) during 1950–1989 (1990–2019) with a weak (strong) NAO-like pattern located near 90°W (30°W) (Fig. 3h–k). In the above, the winter meridional Z500 anomaly dipole over the North Atlantic related to the ENSO or AIV or interannual BKS SIC (Fig. 3g–l) has been defined as a winter NAO-like pattern[57] to

distinguish it from individual sub-seasonal NAO events driven by synoptic-scale eddies over the North Atlantic[58], while the winter average of the NAO events can change the winter NAO-like pattern.

A winter Ural blocking (UB) near 60°E, which reflects the winter-mean effect of UB events, also appeared over the Ural-Siberia region during 1990–2019 (Fig. 3k). However, it almost vanishes during 1950–1989, even though the La Niña-related Pacific positive Z500 anomaly can extend to the east side of Ural Mountains. In fact, in recent years La Niña favors stronger UB events due to reduced mean zonal wind over Eurasia[59,60] via stratospheric influence[61]. Moreover, the BKS SIC decrease (Fig. 3g, j) and the positive phase of AIV (Fig. 3i, l) occur in tandem with a combination of the winter NAO⁺-like pattern with winter UB during 1950–1989 and 1990–2019. The winter UB tends to be located near 60°E during 1990–2019, which is more favorable to the winter BKS SIC decline than the eastward-shifted UB[62] during 1950–1989 (Fig. 3g, i). The strong linkage of the NAO-like pattern to the winter ENSO during 1990–2019, accompanied by a typical winter UB near 60°E, is also in agreement with the result of a strengthened negative correlation between ENSO and the interannual NAO-like pattern in a warming climate[63]. Thus, the AIV is more closely linked to the ENSO during 1990–2019 than during 1950–1989 via the NAO-like anomaly pattern (Fig. 3k, l).

Many observational and model studies have shown that ENSO can cause interannual variability in the winter NAO-like pattern or the AIV via the tropospheric and stratospheric pathways[43,44]. The La Niña favors an intensified North Atlantic jet[44] and thus leads to increased Ural blocking events with NAO⁺ events[64], even though Ural blocking mainly results from the decay of the NAO⁺ event[58]. In contrast, the El Niño suppresses such changes. While the winter NAO⁺-like pattern favors BKS SIC decline through enhanced northward Atlantic ocean heat transport[51], the reverse is seen for the winter negative-phase NAO (NAO⁻)-like pattern associated with El Niño. Thus, the ENSO can influence BKS SIC through modulating the NAO-like anomaly pattern. Intrusion of warm and moist air into the BKS is also important for winter BKS SIC decline[25–27], which depends strongly on whether Ural blocking event is concurrent with the NAO⁺ event[19]. When Ural blocking occurs together with NAO⁺ (NAO⁻) events, the intrusion of warm and moist air from the North Atlantic into the BKS is favored (suppressed). Thus, changes in winter intrusion events associated with Ural blocking under the different phases of NAO can lead to variations of the winter SIC over BKS[19,27]. In summary, the ENSO may influence the BKS SIC mainly via two pathways: Northward transport of Atlantic ocean heat modulated by the AIV related to the interannual NAO-like pattern and the change in the winter-mean intrusion of warm and moist air into the BKS. We emphasize that the atmospheric path may have little time lag between AIV and BKS SIC, while the oceanic pathway may have a time lag of 10–15 years, and the time lag analysis based on Fig. 2 seems to suggest a large role of the oceanic pathway[20–22,50–52].

We find that while the positive phase of AIV corresponds to a BKS SIC decline during both 1950–1989 (Fig. 4e) and 1990–2019 (Fig. 4f), the BKS SIC anomaly is stronger during 1990–2019 than during 1950–1989 due to increased AIV variability over the latter period (Fig. 2b). The reverse is seen during the negative phase of AIV with stronger positive SIC anomalies during 1990–2019 than during 1950–1989 again due to increased AIV variability. In addition, we see that La Niña (El Niño) corresponds to a BKS SIC decline (increase) that is stronger during 1990–2019 (Fig. 4c) than during 1950–1989 (Fig. 4b). As ENSO amplitudes strengthen during 1990–2019 (Fig. 2a), its influence on BKS SIC also increases, leading to increased interannual variability in BSK SIC in recent decades. Our findings are also consistent with the results obtained from the maximum covariance analysis that the La Niña, North Atlantic cold-warm-cold SST tripole, and NAO⁺-like dipole with UB are the optimal patterns for the winter BKS SIC decline (Supplementary Fig. 4). Because the role of the Atlantic ocean heat transport related to the NAO-like pattern in BKS SIC

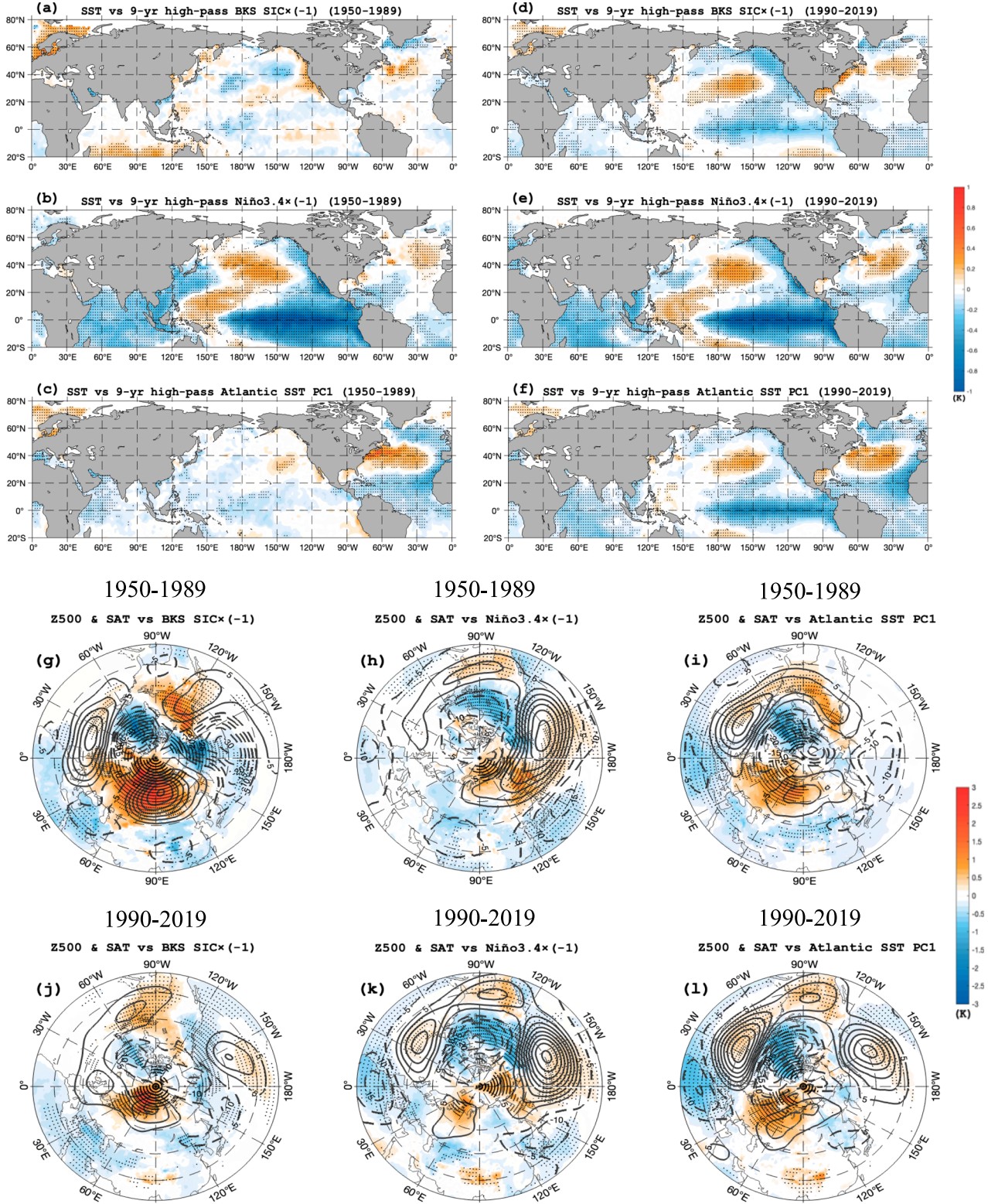

**Fig. 3 | Spatial patterns of winter sea surface temperature (SST), 500-hPa geopotential height (Z500) and surface air temperature (SAT) anomalies related to the interannual variations of the sea ice concentration (SIC) over Barents-Kara Seas (BKS), North Atlantic SST and Niño3.4 index. a–l** Regressed DJF (December- February) mean **a–f** SST and **g–l** Z500 (contour interval = 10, gpm) and SAT (color shading, K) anomalies onto 9-year high-pass filtered normalized **a, d, g, j** BKS SIC, **b, e, h, k** Niño3.4, and **c, f, i, l** North Atlantic SST first principal component (PC1) time series during **a, b, c, g, h, i** 1950–1989 and **d, e, f, j, k, l** 1990–2019. Dotted regions are significant at the 95% confidence level (for color shading) based on a two-sided Student *t*-test.

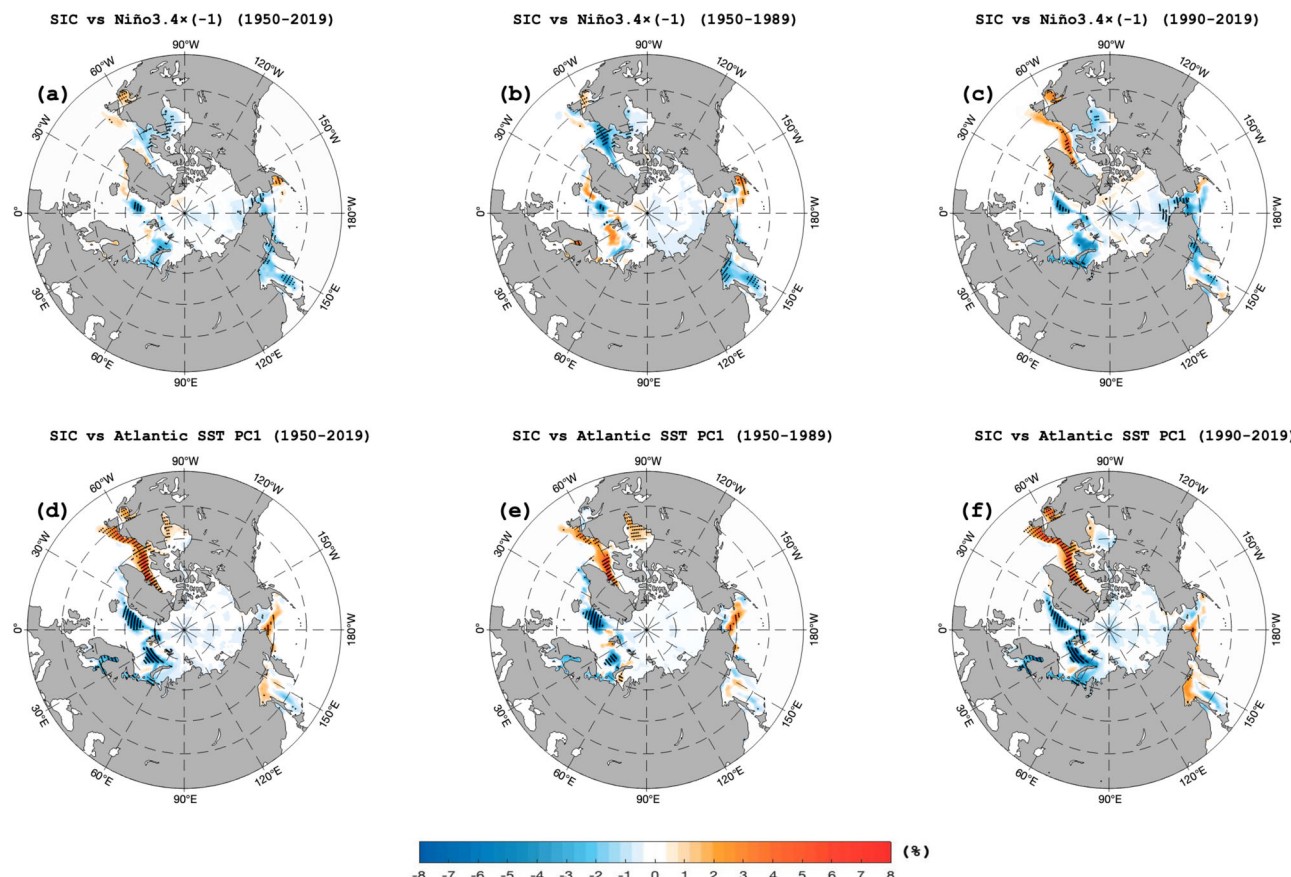

**Fig. 4 | Spatial patterns of winter sea ice concentration (SIC) anomalies in response to the interannual variations of the winter Niño3.4 index and North Atlantic sea surface temperature (SST). a–f** DJF (December-February) mean SIC anomalies (unit: %) regressed onto the 9-year high pass filtered normalized DJF-mean **a–c** Niño3.4 index and **d–f** North Atlantic SST first principal component (PC1) time series during **a, d** 1950–2019, **b, e** 1950–1989 and **c, f** 1990–2019, with the dotted regions being significant ($p < 0.05$) based on a two-sided Student $t$-test.

variations has been widely investigated[20–22,50–53], our analysis below focuses on examining whether the AIV associated with ENSO can induce the interannual variability of the winter BKS SIC through modulating individual UB events. Previous studies indicated that the UB lags the NAO[+] event by 4-7 days and the BKS SIC decline lags the UB by ~4 days[19], while the AIV and associated NAO-like pattern can be considered as a background condition influencing NAO and UB events. Thus, the influence of the atmospheric pathway on the BKS SIC is rapid.

### Role of Ural blocking regulated by Atlantic interannual variability in BKS sea ice variations

Here we define the values ≥0.5 (≤−0.5) of the standard deviation in the normalized 9-year high-pass filtered DJF-mean SST PC1 time series as the positive (negative) phase of AIV, whereas its value between −0.25 and 0.25 is defined as a neutral AIV. Based on this definition, we find 23 positive, 24 negative and 13 neutral AIV winters during 1950–2019. Using the blocking index (Eqs. (2) and (3) in Methods), we also find 36, 36, and 22 UB events for the positive, negative and neutral phases of AIV, corresponding to 1.50, 1.57, and 1.69 blocking events per winter. Clearly, the phase of AIV does not significantly alter the winter UB events, but it can significantly affect the frequency of UB events concurrent with positive, negative and neutral NAO events (referred to as UB-NAO[+], UB-NAO[-] and UB-NAO[0] events) (Fig. 5). During the positive phase of AIV, UB-NAO[+] events are more frequent than UB-NAO[-] and UB-NAO[0] events (left bars in Fig. 5) due to intensified North Atlantic jet[64], but UB-NAO[+] (UB-NAO[-]) events are less (more) frequent during the negative phase of AIV (right bars in Fig. 5) due to reduced North

Atlantic jet. During the neutral phase of AIV, the UB-NAO[+] events have almost the same frequency as that of UB-NAO[-] events (middle bars in Fig. 5). Thus, the positive (negative) phase of AIV favors (inhibits) increased UB-NAO[+] events and decreased UB-NAO[-] events.

Here, we only present the results of time-mean composite daily Z500 and SAT anomalies averaged from lag-10 to 10 days (lag 0 denotes the peak day of blocking) of total UB events in winter (the sum of UB-NAO[+], UB-NAO[-] and UB-NAO[0] events) (Fig. 6) for the negative, neutral and positive phases of AIV. During the positive (negative) phase of AIV, the composite UB pattern is mainly related to the NAO[+] (NAO[-]) events (Fig. 6a, c) and is accompanied by a large decline (increase) of the BKS SIC (Fig. 6d, f) mainly due to enhanced (reduced) downward infrared radiation (IR) (Fig. 6g, i). Such an enhanced (reduced) downward IR is related to increased (decreased) air temperature and water vapor over the BKS (Fig. 6j, l), which plays a major role because the downward sensible and latent heat fluxes as well as the sea ice advection play secondary roles during these events[19,25–27,62]. In fact, increased (decreased) air temperature and water vapor over the BKS are closely related to enhanced (reduced) intrusion of warm and moist air caused by UB-NAO[+] (UB-NAO[−]) events[19]. While the UB during the neutral AIV winter has almost the same amount of water vapor over the BKS as during the positive phase of AIV, the water vapor in warm air over the BKS is mainly located in the lower latitude Arctic during the neutral AIV phase (Fig. 6h) than during the positive AIV phase (Fig. 6i). As a result, the downward IR over the BKS is weak during the neutral AIV phase (Fig. 6k), leading to a relatively weak SIC depletion (Fig. 6e). Thus, the positive (negative) phase of AIV tends to favor the BKS SIC decline (increase) to cause a substantial interannual

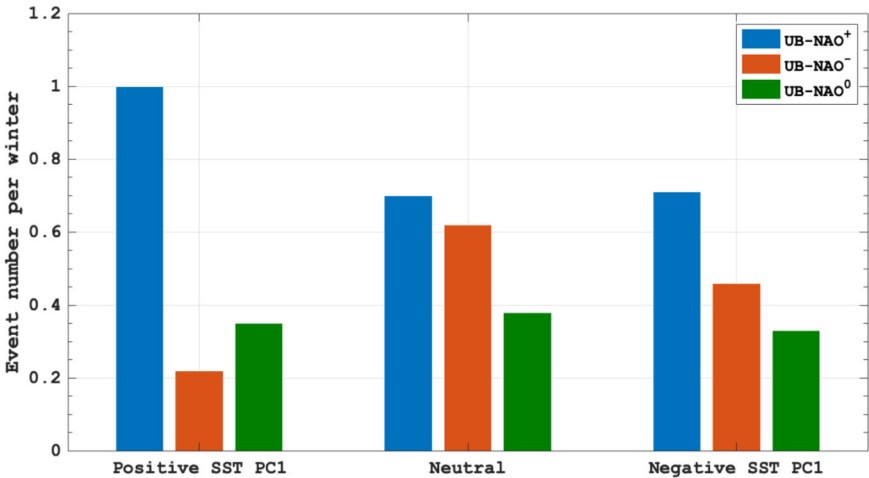

**Fig. 5 | Variations of the event number of winter Ural blocking (UB) events concurrent with the different phases of North Atlantic Oscillation (NAO) with the phase of the Atlantic interannual variability.** The percentage of UB events with three phases of NAO (UB-NAO⁺, UB-NAO⁻ and UB-NAO°) with respect to total UB events for the positive (23 cases, left), neutral (13 cases, middle) and negative (24 cases, right) phases of the 9-year high-pass filtered DJF (December-February) mean North Atlantic SST first principal component (PC1) time series defined as the Atlantic interannual variability.

variability of the BKS SIC via enhanced (reduced) intrusion of warm and moist air and intensified (weakened) surface warming. If the −1.0 (+1.0) standard deviation of the SST PC1 time series is defined as the threshold of a strong negative (positive) AIV winter, the strong positive AIV winter cases exhibit a ~33% increase from 1950–1989 to 1990–2019, whereas the strong negative AIV winter cases show a ~11% increase, thus favoring strong SIC decline and increase events during 1990–2019 (Fig. 1a).

### ENSO as a driver of Atlantic interannual variability via Atlantic Hadley cell variations

The above results reveal that the ENSO and the associated AIV are important for the interannual variability of the BKS SIC. In this section, we show that the AIV is mainly linked to changes in Walker circulation and Atlantic Hadley cell induced by ENSO. We define the difference of domain-averaged 9-year high-pass filtered DJF-mean 500-hPa vertical velocity anomaly between the subtropical North Atlantic (25°N–35°N, 80°–0°W) and equatorial Atlantic (5°S–5°N, 80°–0°W) as a winter Atlantic Hadley cell (AHC) index[65]. We find that the winter Niño3.4 and AHC indices have strong negative simultaneous correlations of −0.76, −0.66 and −0.85 ($p < 0.01$ for all cases) during 1950–2019, 1950–1989, and 1990–2019 (Fig. 7a), respectively. This result suggests that the AHC-ENSO connection is strengthened from 1950–1989 to 1990–2019. We also find that the standard deviation of the Atlantic Hadley cell index increased by ~50% from 0.81 during 1950–1989 to 1.22 during 1990–2019, close to the increase in the AIV. However, how a ~20% increase in ENSO variability leads to the ~50% increase in the variability of the Atlantic Hadley cell or AIV during 1990–2019 is unclear, which may be related to the amplified effect of local air-sea interaction in the North Atlantic[66]. Also note that the Atlantic Hadley cell responds to ENSO with little time lag (Fig. 7a), while Atlantic SST (i.e., AIV) lags the atmospheric NAO-like forcing by ~10 years[67].

Here the winter Walker and Hadley cells are represented by the DJF-mean wind and vertical velocity anomaly fields averaged over 5°S–5°N and over 80°W–0°, respectively[65]. We define 23 El Niño and 20 La Niña winters during 1950–2019 using the Niño3.4 index and the ≥0.5 (≤−0.5) standard deviation thresholds. Composite analysis shows that under La Niña the intensified descending (ascending) branch of the winter Walker cell appears in the equatorial eastern Pacific (equatorial Atlantic) (Fig. 7b), which weakens the mean Walker cell in the equatorial Atlantic and favors an enhanced Atlantic Hadley cell (Fig. 7d). In contrast, the Walker cell shows an opposite change under El Niño

(Fig. 7c), favoring a weakened Atlantic Hadley cell (Fig. 7e). Thus, La Niña (El Niño) leads to intensified (weakened) Atlantic Hadley cell that favors a winter NAO⁺-like (NAO⁻-like) circulation pattern, especially during 1990–2019 as ENSO activity strengthens (Supplementary Fig. 5).

The alternation of the NAO⁺- and NAO⁻-like circulation patterns between La Niña and El Niño and the recent intensification of these patterns can lead to the AIV associated with the interannual variability of AMOC[33,34] that resembles an interannual North Atlantic SST tripole pattern[30]. Moreover, the winter AHC index has a significant positive correlation of 0.3 ($p < 0.05$) with the winter Atlantic SST PC1 time series during 1950–2019, and the correlation increases from −0.09 during 1950–1989 to 0.58 ($p < 0.01$) during 1990–2019, thus suggesting a strengthening of AHC-AIV linkage during 1990–2019. We further see that the NAO⁺-like pattern (for La Niña) of the DJF-mean Z500 anomaly regressed onto the AHC index (Supplementary Fig. 5b) exhibits a striking resemblance to that of the AIV (Fig. 3l). Thus, the AHC-AIV connection is mainly mediated by the variations from NAO⁺-like pattern during La Niña to NAO⁻-like pattern during El Niño. Consequently, the increased ENSO variability in the recent decades (Fig. 2a) can lead to a strengthened AIV (Fig. 2b) via enhanced NAO-like pattern variability[31,37] due to increased variability of the Atlantic Hadley cell (Fig. 7d, e) associated with the change in the Walker cell induced by ENSO (Fig. 7b, c).

### Discussion

In this paper, we found that interannual variability in winter sea ice over the Barents-Kara Seas (BKS) has significantly strengthened from 1950–1989 to 1990–2019 likely due to the increased ENSO amplitude in the recent warm climate. This result has an important implication for improving our understanding of the predictability of the interannual variations of the BKS SIC in recent decades. The pathways of the ENSO's influence on the BKS SIC are illustrated in Fig. 8. During a La Niña winter, the Walker circulation weakens by an anomaly descending (ascending) branch in the tropical eastern Pacific (tropical Atlantic), which leads to an enhanced Atlantic Hadley cell (pink line in Fig. 8) that causes a winter NAO⁺-like anomaly pattern through strengthening the high pressure over its southern lobe. The reverse is seen under El Niño, which strengthens the Walker circulation, leading to a weakened Atlantic Hadley cell that causes a NAO⁻-like anomaly pattern. Under La Niña, a positive-phase Atlantic interannual variability (AIV) associated with a strengthened interannual AMOC, which resembles a cold-warm-

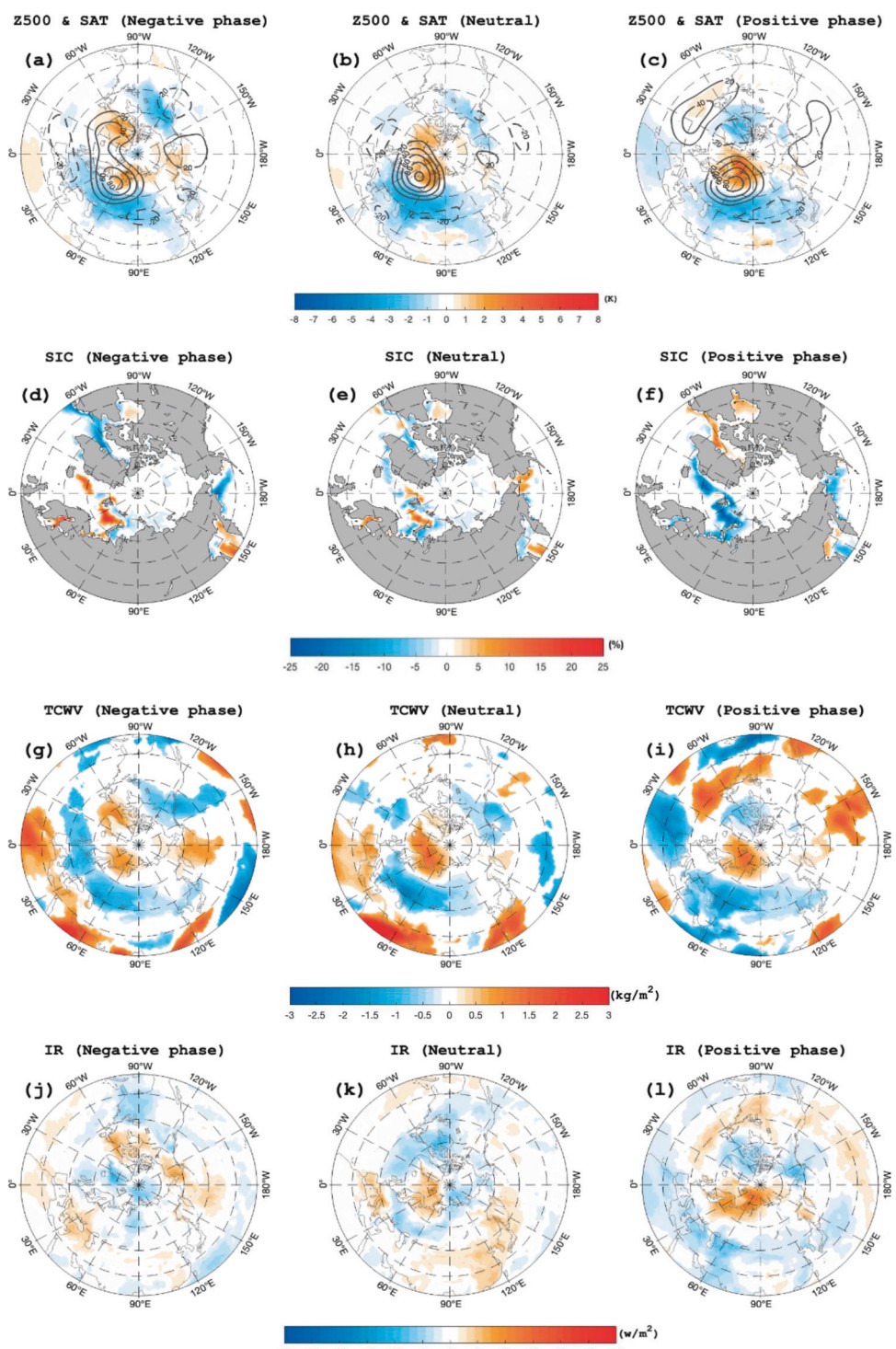

**Fig. 6 | Spatial patterns of composite atmospheric fields and sea ice concentration (SIC) anomaly associated with Ural blocking (UB) events. a–l** Time-mean fields of composite daily **a–c** 500-hPa geopotential height (Z500, contour interval = 20, gpm) and surface air temperature (SAT, color shading, K), **d–f** sea ice concentration (SIC, color shading, %), **g–i** total column water vapor (TCWV, color shading, kg/m²) and **j–l** downward infrared radiation (IR, color shading, w/m²) anomalies averaged from lag −10 to 10 days of UB events for **a, d, g, j** negative, **b, e, h, k** neutral, and **c, f, i, l** positive phases of 9-year high-pass filtered normalized DJF (December-February) mean North Atlantic SST first principal component (PC1) time series during 1950–2019. Lag 0 denotes the peak day of the UB and the color shaded areas are significant at the 5% level based on a two-sided Student *t*-test.

cold SST tripole pattern in the extratropical North Atlantic, can occur as a delayed response to the intensified NAO⁺-like pattern in wind stress or buoyancy forcing[33,34] or the air-sea coupling with the NAO⁺-like pattern[37,68] under La Niña. Similarly, a negative-phase AIV can be generated under El Niño. The positive phase of the AIV is conducive to increased Ural blocking events concurrent with NAO⁺ events through

strengthening the North Atlantic jet[64], whereas the reverse is seen for the negative phase of AIV.

During the positive phase of AIV under La Niña, BKS sea-ice melting can arise from the strengthened transport of Atlantic ocean heat into the BKS (light red line in Fig. 8) due to the NAO⁺-like pattern[51–53] and increased intrusion of warm and moist air associated

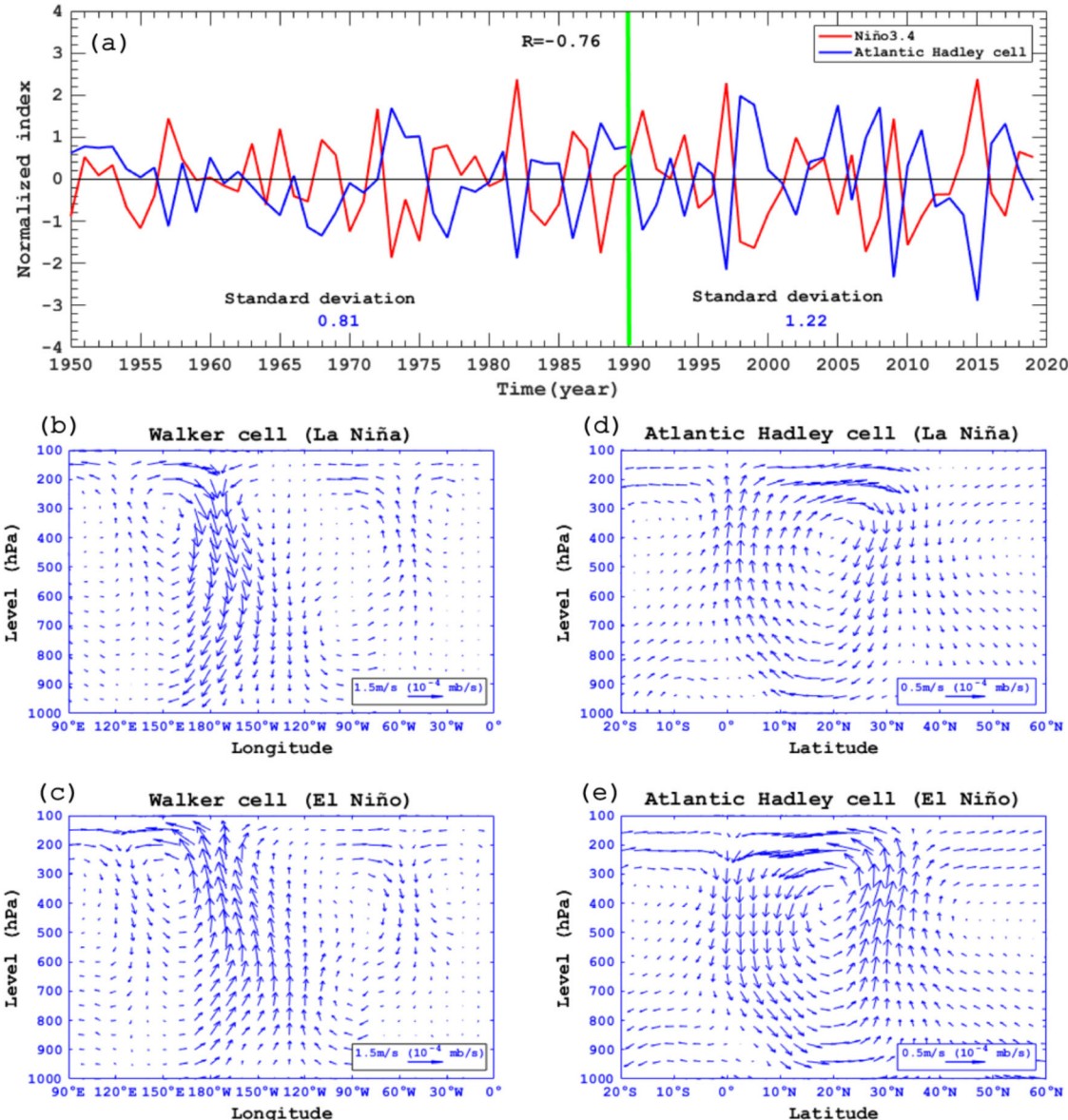

**Fig. 7 | Changes in the winter Walker cell and Atlantic Hadley cell with El Niño and La Niña. a** Time series of normalized DJF (December-February) mean Niño3.4 (red line) and Atlantic Hadley cell (blue line) indices during 1950–2019, where the Atlantic Hadley cell index is defined as the difference of the domain-averaged DJF-mean 500-hPa vertical velocity between the subtropical North Atlantic (25°–35°N, 80°W–0°) and equatorial Atlantic (5°S–5°N, 80°W–0°). The correlation coefficient between the two lines is −0.76 (*p* < 0.01) during 1950–2019. **b, c** Vertical-longitude profiles of the Walker cell defined by DJF-mean zonal wind (m/s) and vertical velocity (mb/s) anomalies averaged over 5°S–5°N for **b** La Niña and **c** El Niño. **d, e** Vertical-latitude profiles of Atlantic Hadley cell defined by DJF-mean meridional wind (m/s) and vertical velocity (mb/s) anomalies averaged over the longitudes 80°W–0° for **d** La Niña and **e** El Niño. The anomaly is relative to the 1950–2019 climatological mean.

with increased Ural blocking events under NAO[+][19]. Increased intrusion accelerates the decline of the winter BKS SIC through increased downward infrared radiation and enhanced surface warming over BKS (green line in Fig. 8)[19,27]. However, the warm air intrusion from the North Atlantic into the BKS is less evident if Ural blocking events are absent[19]. In such a case, the poleward transport of Atlantic ocean heat plays a major role in the BKS SIC decline[51,52]. The reverse is seen for the negative phase of the AIV under El Niño. As ENSO variability is increased from 1950–1989 to 1990–2019, the AIV also strengthens during the later period, leading to increased interannual variability of the BKS SIC during 1990–2019.

It should be mentioned that the mutual relationship between the ENSO and Arctic SIC is complex. Recent model studies suggested that the ENSO might lead to a delayed melting of Arctic SIC in summer[69]. However, the Arctic sea-ice loss may also influence the ENSO, which may depend on the extent of the Arctic sea-ice loss. For example, the results from some models indicated that when Arctic becomes seasonally ice-free, the frequency of strong El Niño events increases significantly, which could happen near the end of the 21st century[70]. Nevertheless, our study reveals that ENSO's significant influence on the interannual variability of the BKS SIC is likely to have happened in the recent decades. To some extent, the enhanced interannual variability of the recent BKS SIC is a footprint of the increased ENSO variability in a warm climate. It is worth mentioning that the frequency of winter UB events[71] and DJF-mean BKS SIC[72,73] are significantly underestimated in CMIP5[72] and CMIP6[73] models to result in a low correlation with ENSO (Supplementary Fig. 2), while the CMIP6 models can well capture the marked increase in interannual variability in the recent BKS SIC.

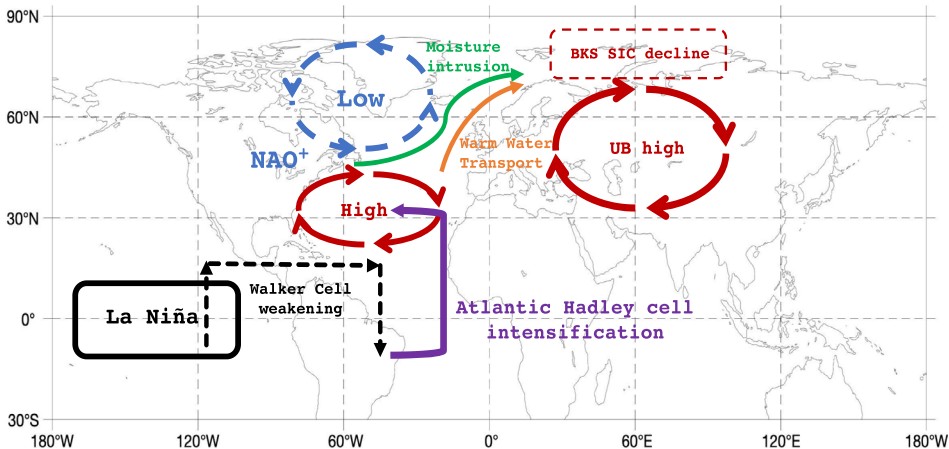

**Fig. 8 | The pathway of El Niño-Southern Oscillation (ENSO) leading to the interannual variability of Barents-Kara sea-ice.** Schematic diagram of La Niña (black box) influencing the decline of sea ice concentration (SIC) over Barents-Kara Seas (BKS) (dashed red box) via the transport of Atlantic warm water as denoted by the Atlantic ocean heat transport (orange red line) and intrusion of warm and moist air (green line) into the BKS. The enhanced northward water transport and air intrusion are related to the positive phase of Atlantic interannual variability (AIV) and the positive North Atlantic Oscillation (NAO⁺)-like anomaly pattern with the Ural blocking (UB) high (anticyclonic red line over Eurasia) related to intensified Atlantic Hadley cell (purple line) and weakened Walker cell (dashed black line). The positive phase of AIV is characterized by the interannual North Atlantic cold-warm-cold sea surface temperature (SST) tripole. The Walker cell weakening refers to an intensified descending (ascending) branch of the winter Walker cell in the equatorial eastern Pacific (equatorial Atlantic). The reversed result is seen for El Niño. The causal sequences can be summarized as follows: ENSO→simultaneous changes in Walker cell and Atlantic Hadley cell→the North Atlantic Oscillation-like anomaly pattern and delayed AIV→ northward intrusion of atmospheric warm air and delayed poleward transport of Atlantic warm water→BKS SIC variations.

In this study, we did not quantify the different roles of the Atlantic warm water transport as defined by the Atlantic ocean heat transport and warm air intrusion in winter BKS SIC variability. Since there are many factors affecting the variability of the BKS SIC, separating the different roles of these factors in the variability of the BKS SIC is difficult because they are often coupled together. This issue is another topic of the future study. On the other hand, the ENSO may influence the BKS SIC via other pathways such as stratospheric pathway[59] because it can influence Ural blocking through the change in the Eurasian tropospheric mean state[61], a poleward shift of the Pacific and Atlantic storm tracks[60], the propagation of ENSO-related teleconnection wave trains[43,44], and eddy-mean flow interaction[74]. These pathways are not considered in this study, which need to be further explored in future.

## Methods

### Empirical Orthogonal Function (EOF) analysis
Because our emphasis is placed on the interannual variability (≤8 years), we use a 9-year high-pass filter to extract the interannual variability. The EOF analysis of the linearly detrended 9-year high-pass filtered DJF-mean SST anomalies over the North Atlantic (80°W–0, 20°–70°N) during 1950–2019 is performed to extract the leading EOF (EOF1) mode of the interannual North Atlantic SST tripole anomaly by computing the eigenvectors of a spatially weighted anomaly covariance matrix. The corresponding PC1 time series of the SST EOF1 mode also reflects the inter-annual variability of Atlantic meridional overturning circulation (AMOC)[37], which is referred to as the Atlantic interannual variability (AIV).

### Power spectrum analysis and wavelet spectrum analysis
We use the power spectrum analysis based on a fast Fourier transform to compute the frequency spectrum of the time series[75]. However, because the time series shows a non-stationary signal, we also use the Morlet wavelet[76] and wavelet coherence spectrum to identify the non-stationary relationship between two time series[47]. The wavelet analysis is a useful method for the identification of periodic signals in the time-frequency space[76], whereas the Wavelet coherence spectrum can well be used to study the time-varying relationship between two time series and their evolution over a continuous time-frequency space by considering the cross-wavelet transform, wavelet power spectrum and phase difference. By calculating the continuous wavelet transform one can well use the wavelet coherence spectrum to identify the relation between two time series (A and B) and their lead-lag relationship. In this wavelet coherence spectrum map, the direction of arrows can reveal the causal relationship between A and B[42]. The significance testing of power spectrum, wavelet spectrum and wavelet coherence spectrum can be found[55,56,76].

### Measure of interannual variability
The standard deviation defined by the square root of the variance of the time series is used as a measure of its interannual variability.

### Regression analysis
The regressed fields of detrended sea surface temperature (SST), 500-hPa geopotential height (Z500) and surface air temperature (SAT) anomalies against 9-yr high pass filtered BKS sea ice and Niño3.4 index as well as SST PC1 time series are derived based on a linear regression equation:

$$Y = \alpha + \beta X, \tag{1}$$

where Y is the predicted variable, the matrix X are independent variables, β is the regression coefficient and α is the error term.

### Identification of blocking events
The one-dimensional blocking index of Tibaldi and Molteni (1990, TM hereafter)[77] is used to identify Ural blocking (UB) events occurring in the Ural Mountains (40°–80°E)[78]. The TM index is defined based on the meridional gradients of 500-hPa geopotential height (Z500):

$$GHGN = [Z500(\phi_N) - Z500(\phi_o)]/(\phi_N - \phi_o), \tag{2}$$

$$GHGS = [Z500(\phi_o) - Z500(\phi_S)]/(\phi_o - \phi_S), \tag{3}$$

at the three given reference latitudes: $\phi_N = 80^oN + \triangle$, $\phi_o = 60^oN + \triangle$, $\phi_S = 40^oN + \triangle$ and $\triangle = -5^o, 0^o, 5^o$ in a fixed longitude. A blocking event is defined to have taken place if the conditions GHGS > 0 and

GHGN < −10 gpm (deg lat)⁻¹ hold for at least three consecutive days and are satisfied for at least one choice of △ in a given zonal region covering at least 15° of longitude.

## Definition of El Niño and La Niña

We define the year with positive (negative) values above (below) 0.5 (−0.5) standard deviations for normalized DJF-mean Niño3.4 index as a positive (negative) phase ENSO or El Niño (La Niña).

## Definition of an individual North Atlantic Oscillation event

We define an individual North Atlantic Oscillation (NAO) event based on the daily NAO index from the NOAA Climate Prediction Center (CPC). A NAO⁺(NAO⁻) event is defined to have taken place if the daily NAO index is above 0.5 (below −0.5) standard deviations and persists for at least three consecutive days[19]. All other NAO events are defined as neutral NAO (NAO°) events. The life cycle of the NAO⁺(NAO⁻) is defined when the daily index starts from a zero value, continues to its peak of the positive (negative) daily NAO index, and then decreases (increases) to the day with a zero value again when the event is ended.

## Definition of Ural blocking events associated with the different phases of NAO

A UB event is defined to be related to an NAO⁺(NAO⁻) event if the peak day of the GHGS occurs within the life cycle of an NAO⁺(NAO⁻) event, which is referred to as a UB-NAO⁺(UB-NAO⁻) event[19]. Similarly, a UB related to a NAO° occurrence is referred to as the UB-NAO° event.

## Data treatment and statistical significance test

All the daily and monthly data were converted into anomaly field by removing the 1950−2019 mean of each calendar day and then de-trended prior to analyses. The sea ice over BKS, Niño3.4 and Atlantic SST EOF PC1 time series with a 9-year high pass smoothing represent their interannual variations. We used a student's $t$-test to examine the statistical significance of the anomaly field and the correlation coefficient in this paper. This statistical significance testing method can be found in the previous work[79]. The 90%, 95 and 99% are denoted by $p < 0.1$, $p < 0.05$ and $p < 0.01$ in turn.

## Data availability

The daily and monthly ERA5 reanalysis data sets used in this paper are available from the ECMWF website (https://cds.climate.copernicus.eu/#!/search?text=ERA5), which include daily 500-hPa geopotential height (Z500) and surface air temperature (SAT) (2 m on the surface of earth). The monthly sea surface temperature (SST) and sea ice concentration (SIC) data are taken from the Hadley Centre (https://www.metoffice.gov.uk/hadobs/index.html). For the DJF-mean SIC, we used the monthly SIC data from the Hadley Centre. The winter Niño3.4 index is taken from the Koninklijk Nederlands Meteorologisch Instituut (KNMI) Climate Explorer (http://climexp.knmi.nl/selectindex.cgi?id=someone@somewhere).

The daily NAO index is taken from the NOAA Climate Prediction Center (CPC) (https://www.cpc.ncep.noaa.gov/products/precip/CWlink/pna/nao.shtml).

We use CMIP6 simulations from 34 Earth System Models for historical (1850−2015) time frame shown in supplementary Table 1 (https://esgf-node.llnl.gov/projects/cmip6/). The monthly-mean SIC data is taken from the CMIP6 models.

## Code availability

The code of the wavelet analysis used in this paper is available at http://atoc.colorado.edu/research/wavelets/. Other codes used in the present study are available from the corresponding author on request.

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

## Acknowledgements

B.L., D.L., and Y.Y. were supported by the National Natural Science Foundation of China (Grant numbers: 42288101, 42150204, and 41975068) and the Chinese Academy of Sciences Strategic Priority Research Program (Grant number: XDA19070403). L.W. is supported by the National Key Research and Development Program of China (2022YFE0106900). A.D. is supported by the National Science Foundation (AGS-1353740 and OISE-1743738). C.X. is supported by the International Cooperation Team on Key Processes and Impacts of the Arctic Cryosphere (BNU 2022-GJTD-01).

## Author contributions

B.L. conceived this study, performed data analysis and plotted all figures besides Fig.7 and wrote this paper in part. D.L. conceived and supervised this study and revised the preliminary manuscript, Y.G. plotted Fig. 7. A.D. improved the manuscript significantly. L.W., I.S., C.X., L.X.W., and Y.Y. participated in constructive discussion and improving the manuscript. B.L., D.L., Y.G., A.D., L.W., I.S., C.X., L.X.W., and Y.Y. contributed to the final version of the manuscript.

## Competing interests

The authors declare no competing interests.
