## [Peer Review File · Nature Communications]

Peer Review Comments, first round review –

Reviewer #1 (Remarks to the Author):

Luo et al. pointed out that on interannual timescale, ENSO can modulate BKL SIC in winter via the affecting AIV (the Atlantic pathway). During the positive phase of AIV, the NAO-like circulation, synergistic with the Ural blocking, promotes the SIC decline over BKS through enhanced poleward transport of Atlantic warm water and atmospheric moisture.

The manuscript is easy to follow and very interesting. These new findings are meaningful to understanding the origin of the interannual variability of winter BKS SIC and provide the theoretical foundation for predicting the Arctic sea ice change and associated climate anomalies. However, more evidence should be provided to make the conclusions robust and the physical mechanisms need further discussion and refinement. Here my concerns and comments:

1. L116-122:

The author pointed out that the AIV is mainly produced by ENSO, but the negative correlation is not very high (-0.28) though passing through 95% confidence level. How much can ENSO explain the interannual variation of AIV? Much more evidence (or relevant references) should be provided to confirm ENSO is the main factor to drive AIV on the interannual timescale. The author can also examine whether the prevailing interannual period of AIV, ENSO and BKS SIC is approximately consistent.

In addition, the correlation of ENSO-BKS SIC is low and insignificant (0.18), with similar conditions occurring in the AIV-ENSO and AIV-BKS SIC correlation. Will the datasets before 1979 with low reliability affect the results? The author should examine the above correlations based on much more datasets for further confirmation.

Does the correlations of three variables are stable? How relevant are them after 1979? The enhanced correlations under global warming or AA would make the manuscript more meaningful. The above correlations are calculated from the reanalysis, whether CMIP6/CMIP5 can reproduce the observed results? Giving more evidence, such as model results, can help improve the credibility of conclusions.

2. L118, L126: Opposite sign of significant correlations between AIV and BKS SIC by using different methods? Please examine it.

3. L134: In recent years, the La Nina-related winter anticyclonic anomaly becomes stronger over the Ural-Siberia (Luo et al., 2021; Zhang et al., 2022). I think the indirect influences of ENSO on BKS SIC may be modulated by distinct atmospheric backgrounds. Can the author give comments on it?

Luo, B., Luo, D., Dai, A., Simmonds, I., & Wu, L. (2021). A connection of winter Eurasian cold anomaly to the modulation of Ural blocking by ENSO. *Geophysical Research Letters*, 48(17), e2021GL094304.

Zhang, X., Wu, B., & Ding, S. (2022). Combined effects of La Niña events and Arctic tropospheric warming on the winter North Pacific storm track. *Climate Dynamics*, 1-18.

4. L135-137: Besides the less strong UB anticyclonic, the anomalous center of NAO shifts toward the eastern North Atlantic, which weakens the moisture intrusion from the North Atlantic to BKS.

5. L173: The UB with NAO- during the negative phase of AIV also enhances the total column (Fig.3e) but with reduced downward IR over BKS. Can the author explain it? Will other variables affect the downward IR?

6. L185-188: The occurrence of UB on the intra-seasonal timescale is crucial for the NAO affecting the BKS SIC. Is UB influenced by ENSO or the internal atmospheric variability? The author should simply examine whether the CMIP6/CMIP5 can obtain the same results.

7. L191-192: After removing the linear effects of ENSO, does the amplitude of AIV significantly attenuate? If not, it is better not to write "mainly results from".

8. L217-221: This manuscript proposes a possible mechanism linking BKS SIC to ENSO through regulating the Atlantic Hadley cell associated with the changed Walker cell. However, the physical processes are very complicated. Except for the stratospheric processes mentioned in this manuscript but are not examined, the tropospheric processes, such as horizontal propagation of planetary waves (PNA) and synoptic eddy-mean flow interaction (North Atlantic) (Ding et al. 2017), are not discussed. Can the author give comments on the role of these mechanisms in the interannual variability of BKS SIC?

Ding, S., W. Chen, J. Feng, HF. Graf (2017), Combined impacts of PDO and two types of La Niña on climate anomalies in Europe. *J. Climate*, 30(9): 3253-3278.

9. Summary: The author should simply emphasize the innovation and significance of this study besides the main conclusion.

10. Figure S2: Can the same situation be observed in different phases of ENSO?

11. The writing needs further improvements for more readable. Some suggestions are shown as follows.

L33, L53, L68: add "the" before "Arctic"

L43: Add a comma before "the enhanced"

L46, L228: change "reversed" to "reverse" or "reversed signal/anomaly"

L52: change "driver" to "driver"

L53: delete "does" and change "influence" to "influences"

L105: change "a" to "an", agree with the beginning sound of the following word SST

L176: According to Figs.3a-b, UB-related Eurasian cold anomaly shows a more eastward rather than westward shift for the positive AIV phase than for its negative phase. Please examine this figure.

L178: delete "the" before "East Asia"

L180: add "more" before "frequent"

L182: Add a comma after "the AIV"

L186: change "phase" to "phases"

Reviewer #2 (Remarks to the Author):

The authors explore links between Atlantic sea surface temperatures and sea ice variability in the Barents-Kara Sea on interannual time scales, ultimately arguing that ENSO impacts sea ice variability in this region through a number of pathways, including the NAO, the local Atlantic sector Hadley cell, and blocking over the Ural Mountains. I found much of the reasoning to be convoluted, and the parts that made clear sense only marginally built on existing studies. In addition, I found the study to be poorly written, with long run on paragraphs and imprecise use of nomenclature. I therefore recommend that the paper be rejected for publication. I've tried to provide constructive comments below to help the authors with future work.

Scientific points

1. All the analysis relies on correlation, yielding no insight into causality. One cannot establish causality this way, and one could equally make the (clearly incorrect) claim that Barents-Kara Sea ice drives ENSO via modulation of sea surface temperatures and the Hadley cell based on the analysis shown in the paper. There is of course evidence that ENSO is the driver, but this comes from other studies. To be constructive, perhaps the authors could further their work through the Granger Causality analysis or some other information theory based approach.

2. The logic is contorted at several points. We are told that ENSO is weakly correlated with Barents-Kara sea ice: $R = 0.18$, suggesting that less than 5% of variance (covariance is proportional to R^2) in sea ice is associated with ENSO. But then the authors argue that this is

because ENSO drives the AIV ($R=-0.24$; hence explaining less than 10% of the variance in sea surface temperature) which in turn drives Barents Kara Sea ice ($R=-0.28$) again explaining less than 10% of the variance. Despite this weak correlation, read the key conclusion of the study, lines 223-6: that ENSO drives Barents Kara Sea ice through SST pattern. Similar convoluted claims are made about the AIV impact sea ice through the NAO, or ENSO and the NAO impacting the ice through Ural Blocking. The final summary Figure 5 did not help me. There appear to be a number of weak pathways, and it's unclear which are most important, or whether they are due to spurious correlation.

3. I'm not convinced that first PCA of 9-year passed Atlantic SST is a coherent mode of variability that deserves the name of "Atlantic Interannual Variability". Why 9 years for the high pass filter? Why the whole North Atlantic? Is this an optimal pattern for explaining Barents-Kara sea ice (to be constructive, consider Maximum Covariance Analysis). And how does the AIV add to our understanding beyond that which could be obtained from the more traditional AMO pattern or the NAO?

4. Moisture transport by the atmosphere and heat transport by the ocean are mentioned at many points (including the grand summary figure 5), but seemingly in passing without proper definition or explanation. It is unclear why moisture transport is more important than sensible heat transport vs. changes in the local energy budget (implied by the analysis of downward IR), or whether thermodynamic impacts are more important than simple advection of sea ice.

Presentation

1. The language is imprecise. It is my understanding that AIV is meant to refer to the first PCA of 9-year high pass Atlantic SSTs (line 111). These two terms are interchanged haphazardly through the manuscript, and as I'll show below, I'm still not sure what they mean. AIV introduced before it is defined, both in the abstract at line 43, and then 89-90, where it is associated with the AMOC, then line 107, and at last finally defined at line 111. Things are further complicated by efforts to take the first PCA of the total Atlantic SST record (which the authors call the AMO), and then high pass filter it (lines 125-126). I understand this produces a similar index, but it's not the same one. Frankly, I still don't understand what AIV is, as at line 118, the Atlantic SST PC1 time series is reported to be correlated at -0.24 with the Barents-Kara sea index, while at line 126, AIV is reported to be 0.38 correlated with Barents-Kara sea index. So are these indices not the same?

2. Long paragraphs make it hard to follow. Consider lines 130-160 which begins by abruptly introducing Ural mountain blocking, wanders through ENSO, and ends with analysis of NAO anomalies. Keeping ideas compartmentalized in paragraphs would help the reader.

3. The paper introduces needless complexity through a series of acronyms, many non-standard. Articles for Nature Communications should be written for a broad scientific audience. As an atmospheric scientist, I am familiar with the NAO, ENSO, and SIC, but was already struggling in the first paragraph where I had to digest BKS and AIV. We are soon barraged with AA (which is often associated with Alcoholics Anonymous) and AMO. Further along, STD (for standard deviation) is introduced but used a total of 4 times (twice to define it); this acronym also carries a less pleasant meaning.

4. Vague language is used at many points, e.g., at line 68 "Atlantic heat or warm water", or line 107 "linked to ENSO probably through the AIV", as are adjectives like "mainly" etc., which lack quantitative rigor.

5. I appreciate that English is not the native language of the lead authors of the study, and that the English use of definite vs. indefinite articles is particularly complicated. An English editor, or help from co-authors more fluent in the language, would help the clarity of the manuscript.

Reviewer #3 (Remarks to the Author):

Review comments on the submitted article "Origins of the interannual variability of sea ice over the Barents-Kara Seas: Atlantic pathway of the ENSO modulation" by B. Luo, D. Luo, Y. Ge, A. Dai, L. Wang, I. Simmonds and L. Wu

General comments

The authors investigate the causes of the interannual variability in Sea-Ice Concentration (SIC) over the Barents-Kara Seas (BKS) by implementing a 9-year high-pass filter to various fields with

view to isolate the signal induced by ENSO variability. Although that the study is based solely on the analysis of statistical relationships and does not include any aspects of a modelling approach, I reckon that the work makes a very useful contribution to the investigation of the remote ENSO influence via the Atlantic atmospheric pathway. My main concern is that at many places the writing style is awkward to the extent that affects the discussion and significantly hinders the interpretation of some key results. This is particularly obvious in the section discussing the role of Ural blocking in tandem with the AIV. As a result, the manuscript needs to undergo extensive improvements in its writing style and presentation of results before it is considered for publication by Nature Communications. Also, the scientific merit of the work will be augmented in the case that the authors discuss the limitations of their work within the framework of other possible tropospheric or stratospheric pathways. Below, I indicate the parts of the manuscript where the discussion needs to be improved.

Major Comments

1) L101-104 & Fig. 1b: The expression "interannual decline of the BKS SIC" is confusing and hinders the interpretation of Fig. 1b. Please rephrase so as to better convey the message that (on interannual time scales) the SIC over the BKS depletes during a La Niña-like SST anomaly over Pacific basin and a cold-warm-cold SST tripole pattern in the North Atlantic mid-high latitudes.

2) L122-129: Here the authors present similar results but this time using unfiltered data. The discussion here should be a bit more detailed and explain the difference to what has been done so far and how the results has changed.

3) Section "Role of Ural blocking regulated by Atlantic interannual variability in the BKS sea ice change."

This section should be significantly revamped, there are too many instances where the writing style is loose and the discussion of results needs to be improved:

i) L169: Do the authors imply that "Composite analysis shows ..."?

ii) L169: "UB mainly occurs ...". This expression gives a sense of frequency in the occurrence of UB with respect to NAO or AIV. However, in Fig. 3ab, only time-mean composites of Z500 & SAT are presented. Do you imply that the UB anomaly is stronger during the positive phase of AIV? Please rephrase and clarify.

iii) L169-170: "... the positive (negative) NAO phase or NAO+ (NAO-) during the positive (negative) phase of AIV...". The sentence is confusing, it needs to be simplified and the use of too many parentheses should be avoided. What I do not understand is that while in Fig. 3ad, the time-mean composites of Z500 & SAT for the positive and negative AIV phase are presented, in the above sentence the expression "during the positive (negative) phase of AIV". This adds another layer of classification with respect to NAO and AIV, which is not what is shown in Fig. 3 and described in the caption.

iv) L175-177: "We also note that the UB-related Eurasian cold anomaly shows a more westward shift for the positive phase of AIV than for its negative phase". I do not see and westward shift but only an eastward shift during AIV+ in Fig. 3b.

v) L181-183: "But during the negative phase winter of the AIV the UB-NAO+ events have the almost same magnitude frequency...".

Do you refer again to the frequency of UB or the intensity/amplitude of the UB anomalies?

vi) L185-186: Given the above-mentioned problems the results concerning the classification of UB occurrence or intensity with respect to NAO and AIV should be carefully rephrased so as they are assessed during a possible second revision.

4) L204-208 & Fig. 4. The results presented here are interesting but again the discussion should be improved.

i) Perhaps I miss something, but how are the fields of divergent horizontal and vertical velocity calculated? Do they correspond to some kind of normalized anomalies with respect to climate? Please clarify in the text and in the caption.

ii) L203-212: Please avoid the excessive use of parentheses in the text that hinders the interpretation of results.

iii) L214-217: This sentence is confusing (also too many parentheses where the p-values are explained). Again, unfiltered data is introduced but the methodology should be explained in a more careful and detailed manner.

5) L246-250: Summary

In this work the remote impact of ENSO on the interannual variability of SIC over the BKS region via the Atlantic path is presented. I particularly like the schematic in Fig. 5. However, the authors should enhance the discussion by making comments concerning the remote impact of ENSO via the stratospheric pathway or even through influencing the mid-latitude wave-guide over the Pacific that could be later communicated over the North Atlantic basin. In other words, the work would be more complete in the case that the authors discuss in more details the limitations of their work within the framework of other possible tropospheric or stratospheric pathways.

Minor comments

1) L51: "on trend" is a bit awkward. Please rephrase.

2) L52: "main driver"

3) L55: "over the Northern Hemisphere continents"?

4) L55: The expression "decline trend" is confusing. Please rephrase.

5) L122: Again, the loose use of "decline" is confusing. Perhaps it should be replaced by "We also note that the BKS SIC depletion (on interannual timescales) occurs in tandem with a combination...".

6) L242: "...remote modulation of ENSO in the Pacific basin..." Do you mean remote modulation of the SIC anomalies by ENSO? If yes, please correct throughout the manuscript.

Response to Reviewer #1's comments for the manuscript (NCOMMS-22-22227)

The authors would like to thank Reviewer #1 for his/her helpful comments. We have made a major revision. The point-to-point responses to Reviewer #1's comments are given below:

Response to general statement:

Question:

Luo et al. pointed out that on interannual timescale, ENSO can modulate BKL SIC in winter via the affecting AIV (the Atlantic pathway). During the positive phase of AIV, the NAO-like circulation, synergistic with the Ural blocking, promotes the SIC decline over BKS through enhanced poleward transport of Atlantic warm water and atmospheric moisture.

The manuscript is easy to follow and very interesting. These new findings are meaningful to understanding the origin of the interannual variability of winter BKS SIC and provide the theoretical foundation for predicting the Arctic sea ice change and associated climate anomalies. However, more evidence should be provided to make the conclusions robust and the physical mechanisms need further discussion and refinement. Here my concerns and comments:

Response:

Thanks for your very encouraging comments and useful suggestions in connection with our manuscript.

According to the three reviewer's comments, we have made a major revision and provided more references and evidence to support our findings and associated physical mechanism. In this revised manuscript, we present an important finding that the winter SIC over Barents-Kara Seas (BKS) shows a notably increased interannual variability from 1950-1989 to 1990-2019, which is likely due to the increased variability of the ENSO amplitude in a warm climate since 1990. This new finding will help improve our understanding of the predictability of the BKS SIC.

Response to major comments:

Question (1)

The author pointed out that the AIV is mainly produced by ENSO, but the negative

correlation is not very high (-0.28) though passing through 95% confidence level. How much can ENSO explain the interannual variation of AIV? Much more evidence (or relevant references) should be provided to confirm ENSO is the main factor to drive AIV on the interannual timescale. The author can also examine whether the prevailing interannual period of AIV, ENSO and BKS SIC is approximately consistent.

In addition, the correlation of ENSO-BKS SIC is low and insignificant (0.18), with similar conditions occurring in the AIV-ENSO and AIV-BKS SIC correlation. Will the datasets before 1979 with low reliability affect the results? The author should examine the above correlations based on much more datasets for further confirmation.

Does the correlations of three variables are stable? How relevant are them after 1979? The enhanced correlations under global warming or AA would make the manuscript more meaningful.

The above correlations are calculated from the reanalysis, whether CMIP6/CMIP5 can reproduce the observed results? Giving more evidence, such as model results, can help improve the credibility of conclusions.

Response:

Thanks for your useful comments. Your suggestions are very good.

In this revised manuscript, we have made a major revision and rewritten the manuscript. Our main results in this revised manuscript are based on dividing 1950-2019 into two sub-periods: 1950-1989 and 1990-2019 because a warm climate mainly occurred after 1990 (Nyenzi and Lefale, 2006, *Advances in Geosciences*, 6, 95–101). For example, observational and model studies have revealed that the ENSO amplitude shows an increased interannual variability in a warm climate due to increased greenhouse gases (GHG), which have been widely noted from observations (Nyenzi and Lefale 2006; Latif, M. and Keenlyside, *PNAS*, 2009) and models (Cai et al., *Nature Climate Change*, 2014, 2015; Cai et al., *Nature Rev. Earth Environ.*, 2021). In other words, extreme (large amplitude) El Niño and La Niña events occurred more frequently during 1990-2019 likely due to increased GHG concentrations, in agreement with the model results of Cai et al. (2014, 2015). Thus, it is reasonable to divide the 1950-2019 period into the two sub-periods of 1950-1989 and 1990-2019. During 1990-2019, the correlations of the three variables are stable and enhanced likely due to the effect of warm climate. Please see brief discussions in **lines 107-121, 130-135 and 147-150** in our

revised manuscript

Question “The author pointed out that the AIV is mainly produced by ENSO, but the negative correlation is not very high (-0.28) though passing through 95% confidence level. How much can ENSO explain the interannual variation of AIV?”

Response:

While the AIV has a negative correlation of -0.28 ($p < 0.1$) with the ENSO during 1950-2019, their correlation increases to -0.51 ($p < 0.01$) during 1990-2019, from -0.049 during 1950-1989. Clearly, the ENSO-AIV connection is strong (weak) during 1990-2019 (1950-1989). That is to say, the correlation of -0.28 during 1950-2019 is mainly influenced by the associations during 1990-2019. Clearly, the AIV and ENSO have a non-stationary relationship. These correlations imply that ENSO explains only 8% of the variance of the AIV during 1950-2019, but accounts for 26% of the variance of the AIV during 1990-2019. Please see brief discussions in **lines 127-135** in our revised manuscript.

Question “Much more evidence (or relevant references) should be provided to confirm ENSO is the main factor to drive AIV on the interannual timescale”.

Response:

In the revised manuscript, we have rewritten our manuscript and provide additional evidence to support our result that the ENSO is the main factor driving the AIV. In particular, the regressed DJF-mean Z500 and SAT anomalies against the DJF-mean Niño3.4 time series during 1950-1989 and 1990-2019 are shown in Fig. A1 (also see Fig. 3 in our revised manuscript).

Figure A1. Regressed DJF-mean Z500 & SAT anomalies onto the 9-yr high-pass Niño3.4 index time series during (a) 1950-1989 and (b) 1990-2019.

Fig. A1 shows that during 1950-1989 (1990-2019), the negative phase of Niño3.4 index or La Niña corresponds to a weak (strong) positive-phase NAO (NAO⁺)-like pattern. Clearly, because the center of the NAO⁺-like pattern is located near 90°W (30°W) during 1950-1989 (1990-2019) (Fig. A1a-b), it is apparent that La Niña corresponds to a weak (strong) NAO⁺-like pattern during 1950-1989 (1990-2019). The negative-phase NAO (NAO⁻)-like pattern is intensified for El Niño during 1990-2019. Through the intensified wind stress forcing and turbulent heat flux anomalies associated with the enhanced NAO-like pattern, the strengthened AIV associated with the interannual AMOC, which resembles an enhanced cold-warm-cold SST tripole pattern, can be generated, as indicated by Smith and Polvani (2021) using a CESM model. Only a strong typical NAO-like pattern can produce a substantial interannual variability of the North Atlantic SST tripole associated with the interannual AMOC (Hakkinen 1999; Zhao et al. 2014, 2017; Smith and Polvani 2021). This is a main cause of why the ENSO-AIV connection is strong during 1990-2019.

Because the NAO-like pattern and SST anomalies associated with the interannual AMOC are strongly coupled, the first EOF (EOF1) mode of the 9-yr high-pass North Atlantic SST anomalies can be used to represent the interannual variability of the North Atlantic SST tripole mode coupled with the interannual AMOC, even though the EOF1 mode of the 10-yr low-pass North Atlantic SST anomalies can represent the interdecadal AMOC. In particular, of considerable importance here, Smith and Polvani (2021) found that the Atlantic interannual variability is more strongly influenced

by the ENSO-NAO teleconnection than the NAO itself. In the Introduction and the text of the manuscript, we have added some references and provide more evidence to explain how ENSO drives the interannual variation of AIV via changes in the Atlantic Hadley cell and NAO-like patterns (also see **lines 84-91 and 136-144** in our revised manuscript). For example, because the positive phase of AIV corresponds to an NAO⁺-like pattern (Fig. 3i, l) and a cold-warm-cold SST tripole (Fig. 3c, f), and because an enhanced Atlantic Hadley cell corresponds to a strengthened NAO⁺-like pattern (**Supplementary Fig. 5**), the ENSO can drive the AIV through the adjustment of oceanic Rossby waves and mixed layer to wind stress forcing and turbulent heat fluxes under changes in the Atlantic Hadley cell and NAO-like pattern. This physical mechanism has been described in **lines 322-333** in our revised manuscript.

Question “The author can also examine whether the prevailing interannual period of AIV, ENSO and BKS SIC is approximately consistent”.

Response:

In our revised submission (**lines 158-172**), we have added a wavelet power spectrum map (Fig. 2) which indicates the overall consistency of the prevailing interannual variations of AIV, ENSO and BKS SIC. It is found that the AIV, ENSO and BKS SIC are strongly correlated during 1990-2019, which are especially evident in a warmer climate.

Question “the correlation of ENSO-BKS SIC is low and insignificant (0.18), with similar conditions occurring in the AIV-ENSO and AIV-BKS SIC correlation. Will the datasets before 1979 with low reliability affect the results? The author should examine the above correlations based on much more datasets for further confirmation.”

In this revised manuscript, we have pointed out that the low correlation between ENSO and BKS SIC can be seen as due to a weak ENSO-BKS SIC linkage during 1950-1989. While the correlation between ENSO and BKS SIC is 0.18 ($p>0.1$) during 1950-2019, its correlation is only -0.01 during 1950-1989 and 0.32 ($p<0.05$) during 1990-2019. This suggests that there is a strong ENSO-BKS SIC connection during 1990-2019. This is easily explained. As shown in Fig. A1, La Niña corresponds to a weak (strong) NAO⁺-like pattern without (with) Ural blocking during 1950-1989 (1990-2019). Thus, under a strong NAO⁺-like pattern with Ural blocking as observed during 1990-2019, the Atlantic ocean heat and atmospheric moisture can easily be transported to the

BKS as noted in Schlichtholz (2011), Lien et al. (2017) and Luo et al. (2017). This is why a strong BKS decline can be seen during 1990-2019. This situation is not easily allowed during 1950-1989. As a result, the correlation between the ENSO and BKS SIC is stronger during 1990-1989 than during 1950-1989. Moreover, in the revised manuscript we also find that the AIV-ENSO and AIV-BKS SIC correlations are also stronger during 1990-2019 than during 1950-1989.

In the manuscript, the winter BKS SIC is calculated based on the monthly SIC data from the Hadley Centre. Using the daily ERA5 reanalysis data, we find that the BKS SIC has a correlation of -0.05 (0.23, $p < 0.1$) with the Niño3.4 index during 1950-1989 (1990-2019), which becomes 0.35 (0.74) for 4-8-yr band-pass filtered data. The ERA5 data result is consistent with that of the Hadley center monthly SIC data. Thus, the low correlation before 1990 is reliable. However, the correlation between the ENSO and the interannual variability of the winter BKS SIC based on the 34 CMIP 6 model is very low mainly because these models fail to simulate the interannual variation of the BKS SIC anomalies, NAO and blocking (Li, et al. 2017, Nature comm; Davini et al. 2020, JC; Yamagami et al. 2022, Nature Comm).

Please see brief discussions in **lines 119-121 and 391-395** in our revised manuscript.

Question “Does the correlations of three variables are stable? How relevant are them after 1979? The enhanced correlations under global warming or AA would make the manuscript more meaningful.”

Thanks for this explicit comment which is, in turn, related to the issues addressed above.

In this revised manuscript, we have pointed out that the correlations of ENSO, AIV and BKS SIC are unstable during 1950-2019, which are large during 1990-2019, but small during 1950-1989. We find that their enhanced correlations mainly take place in a warm climate (global warming or AA period) because 1990-2019 can be considered as warm climate decades. We also suggest that the enhanced correlation in a warmer climate is likely due to increased variability of the ENSO amplitude under GHG-induced warming, as demonstrated by Cai et al. (2014, 2015, 2021), for example. They found that extreme (large amplitude) El Niño and La Niña events can more frequently occur as a result of intensified GHG-induced warming. To some extent, the increased variability of the ENSO amplitude can explain the notably increased variability of the winter BKS SIC from 1950-1989 to 1990-2019. In other words, the recent substantial interannual variability of the winter

BKS SIC likely originates from the increased variability of the ENSO amplitude in a warm climate. In this revised manuscript, we have emphasized this new and important result.

Please see brief discussions in **lines 107-121, 127-135, 147-151 and 153-156** in our revised manuscript.

Question “The above correlations are calculated from the reanalysis, whether CMIP6/CMIP5 can reproduce the observed results? Giving more evidence, such as model results, can help improve the credibility of conclusions”.

Response:

This is a good suggestion, and we have performed additional analyses to follow it up. We used the historical (1950-2013) monthly SIC data of 34 models from CMIP6 to calculate the time series of DJF-mean SIC anomaly averaged over BKS and its power spectrum, and show the results in Fig. A2. The results show that the BKS SIC has multi-decadal, decadal and interannual variations, which are well captured by the CMIP6 multi-model ensemble. However, its correlation with the ENSO index is low, namely 0.11 during 1950-1989 but only -0.08 during 1990-2013. Clearly, the CMIP6 multi-model ensemble cannot capture the strong ENSO-BKS SIC linkage in a warm climate. However, the CMIP6 models can well simulate a main feature that the winter BKS SIC shows a marked increase in interannual variability from 1950-1989 to 1990-2013. The CMIP6 result is also shown in Fig. S2. In fact, previous studies have indicated the inability of many climate models in simulating the SIC variability over Arctic. The CMIP6 multi-model ensemble can capture the interannual variability of the winter BKS SIC (Fig. A2c), even though the 9-yr high-pass winter BKS SIC has a low correlation with the ENSO. However, for the 4-8-yr band-pass filtered data, the winter BKS SIC has a correlation of 0.35 (-0.52) with the ENSO index during 1990-2013 (1950-1989). The CMIP6 multi-model ensemble confirms that the winter BKS SIC shows a substantial interannual variability in a warm climate, consistent with the results from the monthly SIC data from the Hadley Centre and ERA5 data. Please see brief discussions in **lines 119-121 and 391-395** in our revised manuscript.

Figure A2. (a) Time series of normalized DJF-mean Arctic sea ice concentration (SIC) anomaly averaged over Barents-Kara Seas (BKS) (30° - 90° E, 65° - 85° N) for detrended (blue line) and 9-years (9-yr) high-pass filtered (black line) CMIP6 model data (34 models), where the blue (red) value in the left- (right-) hand side of the green line represents the mean standard deviations of the BKS SIC variations averaged over 1950-1989 (1990-2013). (b, c) Power spectra of the (b) detrended and (c) 9-yr high-pass filtered BKS SIC time series. In panels b-c, The blue (red) dashed line represents the 90% (95%) confidence level.

Question (2)

2. L118, L126: Opposite sign of significant correlations between AIV and BKS SIC by using different methods? Please examine it.

Response:

Thanks for alerting us to this poor expression. This has now been corrected.

Question (3)

L134: In recent years, the La Niña-related winter anticyclonic anomaly becomes stronger over the Ural-Siberia (Luo et al., 2021; Zhang et al., 2022). I think the indirect influences of

ENSO on BKS SIC may be modulated by distinct atmospheric backgrounds. Can the author give comments on it?

Luo, B., Luo, D., Dai, A., Simmonds, I., & Wu, L. (2021). A connection of winter Eurasian cold anomaly to the modulation of Ural blocking by ENSO. *Geophysical Research Letters*, 48(17), e2021GL094304.

Zhang, X., Wu, B., & Ding, S. (2022). Combined effects of La Niña events and Arctic tropospheric warming on the winter North Pacific storm track. *Climate Dynamics*, 1-18.

Response:

Indeed, in recent years the La Niña-related winter anticyclonic anomaly has become stronger over the Ural-Siberia region (Luo et al., 2021; Zhang et al., 2022) via the stratospheric influence (Butler et al. 2014). In **lines 203-205 and 401-403** of the revised manuscript we have added some discussion on this issue.

Question (4)

L135-137: Besides the less strong UB anticyclonic, the anomalous center of NAO shifts toward the eastern North Atlantic, which weakens the moisture intrusion from the North Atlantic to BKS.

Response:

Because we have rewritten the manuscript, these descriptions have been revised. In fact, the combination of UB with the different phase of NAO is crucial for the moisture intrusion from the North Atlantic to the BKS, as shown in Fig. 6.

Question (5)

L173: The UB with NAO- during the negative phase of AIV also enhances the total column (Fig.3e) but with reduced downward IR over BKS. Can the author explain it? Will other variables affect the downward IR?

Response:

In this revised manuscript, we have pointed out that the negative (positive) phase of AIV favors UB with NAO⁻ (NAO⁺) or UB-NAO⁺ (UB-NAO⁻) events. The UB-NAO⁺ (NAO⁻) event favors (inhibits) the intrusion of water vapor from the North Atlantic to the BKS. This has been discussed in Luo et al. (2017, ERL) in detail. In fact, the content of water vapor mainly determines the downward IR and surface warming, whereas the variability of the water vapor is mainly associated with

atmospheric circulation patterns (Gong and Luo, JC, 2017). The impact of other variables (i.e., sensible and latent heat fluxes) is weak. Thus, in **lines 281-290** of the revised manuscript, we only use the previous results to indicate that increased (decreased) water vapor intrusion is related to UB-NAO⁺ (UB-NAO⁻) events.

Question (6)

L185-188: The occurrence of UB on the intra-seasonal timescale is crucial for the NAO affecting the BKS SIC. Is UB influenced by ENSO or the internal atmospheric variability? The author should simply examine whether the CMIP6/CMIP5 can obtain the same results.

Response:

Our previous studies have revealed that most of UB events mainly result from the decay of NAO⁺ events via energy dispersion (Luo et al. 2016, JC), whereas NAO events are driven by high-frequency (≤ 7 days) synoptic-scale eddies in the North Atlantic (Luo et al. 2007, JAS). Thus, the ENSO is not necessary for the generation of NAO⁺ and UB events with a sub-seasonal timescale (10-20 days). However, ENSO can influence the interannual North Atlantic jet through the interannual NAO⁺-like pattern variability. Essentially speaking, the interannual North Atlantic jet variability can be considered as a background condition affecting the evolution of sub-seasonal UB and NAO⁺ events. As indicated in Luo et al. (2018, JAS), the intensified background North Atlantic jet favors increased UB events through the decay of NAO⁺ events due to intensified energy dispersion. Thus, ENSO can influence the frequency of UB-NAO⁺ or UB-NAO⁻ events through modulating the interannual North Atlantic jet. For example, La Niña (El Niño) corresponds to an intensified (weakened) interannual NAO⁺-like pattern to produce a strong (weak) North Atlantic jet. In this case, La Niña favors UB-NAO⁺ events and suppresses UB-NAO⁻ events, whereas El Niño suppresses UB-NAO⁺ events and favors UB-NAO⁻ events. This is why the positive (negative) phase of AIV or ENSO favors (suppresses) increased UB-NAO⁺ events and decreased UB-NAO⁻ events, as shown in Figure 5 or Fig. A3.

The question as to whether the CMIP models can capture the same structures as exposed in the observations is a good and revealing one. Davini et al. (2020, JC) used the CMIP6 simulations to evaluate the ability of the climate models to simulated blocking and found that from CMIP3 (2007) and CMIP5 (2012) to CMIP6 (2019) all models underestimated blocking frequencies, and this is still true for the CMIP6 models. In particular, the CMIP6 models seriously underestimate the frequency

of Ural blocking events (Davini et al. 2020, JC; Chen et al. 2021, JC), which are particularly relevant to the processes discussed here. The fact that the CMIP models cannot realistically simulate Ural blocking frequency may lead to the low correlations of the CMP6 ensemble DJF-mean BKS SIC with the winter Niño3.4 index (supplementary Fig. 2) (Li et al. 2017; Yamagami et al. 2022). In this revised manuscript, we have now explicitly made this point at **lines 391-395**.

Davini, P. and D. D'Andera, 2020: From CMIP3 to CMIP6: Northern hemisphere atmospheric blocking simulation in present and future climate, *J. Climate*, 33, 10021–10038.

Chen, X., D. Luo, Y. Wu, E. Dunn-Sigouin and J. Lu, 2021: Nonlinear response of atmospheric blocking to early winter Barents-Kara Seas warming: An idealized model study. *J. Climate*, 34, 2367-2383.

Li D., R. Zhang and Knutson, T. R. On the discrepancy between observed and CMIP5 multi-model simulated Barents Sea winter sea ice decline, *Nature Comm.*, 8:14991 (2017)

Yamagami, Y., M. Watanabe, M. Mori and Ono, J. Barents-Kara sea-ice decline attributed to surface warming in the Gulf Stream, *Nature Comm.*, 13:3767 (2022).

Question (7)

L191-192: After removing the linear effects of ENSO, does the amplitude of AIV significantly attenuate? If not, it is better not to write “mainly results from”.

Response:

In this manuscript, we have removed the linear trends of the Niño3.4 index and AIV. Hence the detrending does not influence the interannual variability of the BKS SIC. In this revised version, we have replaced “mainly results from” by “mainly linked to”.

Question (8):

L217-221: This manuscript proposes a possible mechanism linking BKS SIC to ENSO through regulating the Atlantic Hadley cell associated with the changed Walker cell. However, the physical processes are very complicated. Except for the stratospheric processes mentioned in this manuscript but are not examined, the tropospheric processes, such as horizontal propagation of planetary waves (PNA) and synoptic eddy-mean flow interaction (North Atlantic) (Ding et al. 2017), are not discussed. Can the author give comments on the role of these mechanisms in the interannual variability of BKS SIC?

Ding, S., W. Chen, J. Feng, HF. Graf (2017), Combined impacts of PDO and two types of La

Niña on climate anomalies in Europe. J. Climate, 30(9): 3253-3278.

Response:

Thanks for this comment in connection with the complexity of the processes. In the revised manuscript we have added some clarifying discussion regarding the possible roles of stratospheric processes, the teleconnection wave train propagation, poleward shift of Pacific and Atlantic storm tracks and eddy-mean flow interaction (Ding et al. 2017) in the BKS SIC variability in **lines 401-406** of the revised manuscript.

Question (9):

Summary: The author should simply emphasize the innovation and significance of this study besides the main conclusion.

Response: Thanks for commenting on the value of emphasizing the innovation and significance of our investigation. We have followed up this good suggestion with some additional text at **lines 350-354** of the revised manuscript.

Question (10):

Figure S2: Can the same situation be observed in different phases of ENSO?

Response:

Thanks for this interesting question. To look at this we have calculated the numbers of UB-NAO⁺, UB-NAO⁻ and UB-NAO⁰ events for La Niña, Neutral and El Niño winters, and show their results in Fig. A3.

Figure A3. The percentage of UB events with three phases of NAO (UB-NAO⁺, UB-NAO⁻ and UB-NAO⁰) with respect to total UB events for the positive (23 cases, **left**), neutral (13 cases, **middle**) and negative (24 cases, **right**) phases of the 9-yr high-pass Niño3.4 time series. The neutral winter is defined as the value of the Niño3.4 being between -0.5 and 0.5

The figure shows that La Niña favors UB-NAO⁺ events and inhibits the UB-NAO⁻ events, whereas El Niño suppresses UB-NAO⁺ events and favors UB-NAO⁻ events. This conclusion is broadly consistent with the results of AIV.

Question (11):

The writing needs further improvements for more readable. Some suggestions are shown as follows.

L33, L53, L68: add “the” before “Arctic”

L43: Add a comma before “the enhanced”

L46, L228: change “reversed” to “reverse” or “reversed signal/anomaly”

L52: change “deriver” to “driver”

L53: delete “does” and change “influence” to “influences”

L105: change “a” to “an”, agree with the beginning sound of the following word SST

L176: According to Figs.3a-b, UB-related Eurasian cold anomaly shows a more eastward rather than westward shift for the positive AIV phase than for its negative phase. Please examine this figure.

L178: delete “the” before “East Asia”

L180: add “more” before “frequent”

L182: Add a comma after “the AIV”

L186: change “phase” to “phases”

Response:

Thank you for spiting these slips. All of these have now been corrected.

Response to Reviewer #2's comments for the manuscript (NCOMMS-22- 22227)

We appreciate the very helpful comments of Reviewer #2, and have undertaken a major revision in light of them. Our point-to-point responses to the comments are given below:

Response to general statement:

Question:

The authors explore links between Atlantic sea surface temperatures and sea ice variability in the Barents-Kara Sea on interannual time scales, ultimately arguing that ENSO impacts sea ice variability in this region through a number of pathways, including the NAO, the local Atlantic sector Hadley cell, and blocking over the Ural Mountains. I found much of the reasoning to be convoluted, and the parts that made clear sense only marginally built on existing studies. In addition, I found the study to be poorly written, with long run on paragraphs and imprecise use of nomenclature. I therefore recommend that the paper be rejected for publication. I've tried to provide constructive comments below to help the authors with future work.

Response:

Thanks for your useful comments on the improvement of our submission.

In the revised manuscript, we have made a major revision, rewritten and re-organized the manuscript according to your comments. Moreover, in the revised version we have emphasized the non-stationary relationship between the BKS SIC and ENSO. We have further found that the interannual variability of the winter BKS SIC is more closely linked to ENSO during 1990-2019 than during 1950-1989 because their correlation coefficient is -0.01 (0.32) during 1950-1989 (1990-2019), indicating that the ENSO-BKS SIC connection is significantly intensified during 1990-2019. On this basis, it is inferred that the increased interannual variability of the BKS SIC in the recent decades is likely due to an increased variability of the ENSO amplitude in a warm climate.

In the revised manuscript we have made significant improvements in the writing and also reduced the length of many paragraphs. These have certainly made the paper flow much more easily, and also made the underlying reasoning and arguments much more transparent.

Response to Scientific points:

Question (1):

All the analysis relies on correlation, yielding no insight into causality. One cannot establish causality this way, and one could equally make the (clearly incorrect) claim that Barents-Kara Sea ice drives ENSO via modulation of sea surface temperatures and the Hadley cell based on

the analysis shown in the paper. There is of course evidence that ENSO is the driver, but this comes from other studies. To be constructive, perhaps the authors could further their work through the Granger Causality analysis or some other information theory based approach.

Response:

Thanks for your useful comments. We appreciate the important point being made.

To approach this issue, we have investigated the causality between ENSO and AIV or BKS SIC by using Morlet wavelet (Grinsted et al. 2004; Torrence and Webster 1999; Börgel et al. 2020) to calculate the wavelet power and coherence spectra of two time series (X and Y). The wavelet coherence spectrum can be used to detect common time-localized oscillations in nonstationary signals. In situations where it is natural to view one time series as influencing another, one can use the phase and arrow of the wavelet cross-spectrum to identify the phase relation and relative time lag between the two time series (X and Y).

In the wavelet coherence spectrum map, arrows pointing to the right-down or left-up indicate that the first variable (X) is leading, while arrows pointing to the right-up or left-down show that the second variable (Y) is leading. Arrows pointing to the right (left) are seen when the time series are in phase (anti-phase). Thus, we may use the wavelet coherence spectrum map to infer their causality (Torrence and Webster, 1999; Grinsted et al. 2004). It is found from the wavelet power spectra (Fig. B1) of the BKS SIC, Atlantic SST PC1 and Niño3.4 time series that the strong variability of the Niño3.4 index in the 2-6-yr band during 1980-2019 starts earlier than the strong variability of the Atlantic SST PC1 in 4-8-yr and 2-4-yr bands during 1990-2019, while the beginning of the strong SST PC1 variability is earlier than that of the intense BKS SIC variability during 1995-2019. Thus, the ENSO leads the AIV and then leads the BKS SIC. We also see from the wavelet coherence spectra (Figs. B1d-f) that the ENSO leads the AIV and then leads the BKS SIC. Some discussion on such a causality has been added in **lines 158-172**.

Grinsted, A., J. C. Moore, and Jevrejeva, S. Application of the cross wavelet transform and wavelet coherence to geophysical time series. *Nonlinear Proc. Geophys.*, 11, 561–566 (2004).

Torrence, C. and Webster, P. J. Interdecadal Changes in the ENSO–Monsoon System. *J. Climate*, 12, 2679-2690 (1999).

Börgel, F., C. Frauen, T. Neumann and Meier, H. E. M. The Atlantic Multidecadal Oscillation controls

the impact of the North Atlantic Oscillation on North European climate, *Environ. Res. Lett.*, 15 104025 (2020).

Figure B1. (a, b, c) Wavelet power spectra normalized by the variances of winter (a) BKS SIC, (b) Atlantic SST PC1 and (c) Niño3.4 time series during 1950-2019, where the ordinate is the Fourier period (year) and the abscissa is time. The thick contour encloses regions with at least the 95% confidence and the thin line indicates the limit of the cone of influence. (d, e, f) Wavelet coherence spectra of (d) BKS SIC and Niño3.4, (e) SST PC1 with Niño3.4 and (f) BKS SIC and SST PC1. The arrows in the significant regions indicate the phase relationship between the BKS SIC or SST PC1 and Niño3.4 with the in-phase pointing right (antiphase pointing left).

Question (2)

The logic is contorted at several points. We are told that ENSO is weakly correlated with Barents-Kara sea ice: $R = 0.18$, suggesting that less than 5% of variance (covariance is proportional to R^2) in sea ice is associated with ENSO. But then the authors argue that this is because ENSO drives the AIV ($R = -0.24$; hence explaining less than 10% of the variance in sea

surface temperature) which in turn drives Barents Kara Sea ice ($R=-0.28$) again explaining less than 10% of the variance. Despite this weak correlation, read the key conclusion of the study, lines 223-6: that ENSO drives Barents Kara Sea ice through SST pattern. Similar convoluted claims are made about the AIV impact sea ice through the NAO, or ENSO and the NAO impacting the ice through Ural Blocking. The final summary Figure 5 did not help me. There appear to be a number of weak pathways, and it's unclear which are most important, or whether they are due to spurious correlation.

Response:

Thanks for these comments, and we agree that we had presented our arguments poorly. We have addressed this (and many other points) in a major revision of the manuscript. On this specific point, we found the relationship between the ENSO and BKS SIC to be non-stationary. While the ENSO is weakly correlated ($R=0.18$) with the BKS SIC during the entire period of 1950-2019, their correlation is much larger during 1990-2019. We note that the correlation between the ENSO and BKS SIC is -0.01 during 1950-1989, but it becomes 0.32 ($p<0.05$) during 1990-2019, thus explaining that 10% of the variance in the BKS SIC is linked to ENSO during 1990-2019. For 4-8-yr band pass filtered data, their correlation coefficient becomes 0.2 (0.63) during 1950-1989 (1990-2019), thus indicating that 4% (40%) of the variance in the BKS SIC is associated with ENSO during 1950-1989 (1990-2019) for 4-8-yr band pass filtered data. Indeed, the AIV has only a negative correlation of -0.28 with ENSO. But their correlation becomes -0.049 (-0.51) during 1950-1989 (1990-2019), thus explaining that 0.2% (26%) of the variance in the AIV is related to ENSO during 1950-1989 (1990-2019). This demonstrates how the ENSO-AIV connection is significantly strengthened from 1950-1989 to 1990-2019. The AIV-BKS SIC linkage is also significantly intensified from 1950-1989 to 1990-2019.

In the revised manuscript, we have moved our focus to explain why the ENSO-BKS SIC connection is intensified during 1990-2019 and provided an explanation that the increased variability of the ENSO amplitude in a warmed climate since 1990s (Nyenzi and Lefale 2005; Cai et al. 2014, 2015, 2021) is likely an important factor influencing the substantial interannual variability of the recent BKS SIC. We have also explained in detail the pathway of ENSO influencing the BKS SIC in Figure 8 as a schematic diagram, and concluded that this pathway is strong in a warm climate. Thus, the ENSO is a primary source of the interannual variability of the recent BKS SIC via the pathway from Atlantic Hadley cell to NAO-like pattern/AIV and then to the BKS SIC due to poleward transports of

Atlantic ocean heat and atmospheric moisture.

Question (3)

I'm not convinced that first PCA of 9-year passed Atlantic SST is a coherent mode of variability that deserves the name of "Atlantic Interannual Variability". Why 9 years for the high pass filter? Why the whole North Atlantic? Is this an optimal pattern for explaining Barents-Kara sea ice (to be constructive, consider Maximum Covariance Analysis). And how does the AIV add to our understanding beyond that which could be obtained from the more traditional AMO pattern or the NAO?

Response:

Thanks for these comments.

We note in our revised manuscript that, because our study is focused on the influence of ENSO on the BKS SIC, we only consider the interannual variability with the timescale ≤ 8 years. Thus, it is reasonable to use a 9-year (9-yr) high-pass filter to remove decadal and multi-decadal variations (≥ 9 years) to retain the interannual variability of the North Atlantic SST tripole anomalies. In previous studies, it has been recognized that the North Atlantic SST tripole possesses a notable decadal and interannual variability (e.g., Czaja and Marshall, 2002). Thus, here we use a 9-yr high-pass filter to extract the interannual variability of the North Atlantic SST tripole and define it as the Atlantic interannual variability (AIV). Such a new definition can reflect the interannual variability of the North Atlantic SST tripole to distinguish the decadal part of the North Atlantic SST tripole. In this connection we have pointed out that the Atlantic interannual variability is a coherent mode associated with the interannual NAO-like pattern, as seen from the maximum covariance analysis below. In our revised manuscript, we used the North Atlantic domain (20°-70°N, 80°W-0°) rather than the whole North Atlantic.

To further examine the sensitivity of the AIV time series to the timescale of the high-pass filter and the size of the North Atlantic domain. The EOF1 and PC1 time series of 10-yr high-pass filtered DJF-mean SST anomaly over the 'North Atlantic', defined here as 20°-70°N, 80°W-0° and 9-yr high-pass filtered DJF-mean SST anomaly when the North Atlantic is delimited as 10°-70°N, 80°W-0°, are shown in Fig. B2. It is clear that the 10-yr high-pass PC1 time series has a very strong correlation with that of the 9-yr high pass PC1 time series. If one uses the domain (10°-70°N, 80°W-0°), the PC1 has a correlation of 0.84 with the PC1 time series over the smaller domain (20°-70°N, 80°W-0°). Thus,

although using a 10-yr high pass filter has the same result as using a 9-yr high-pass filter, the result is not strongly influenced by a change in the domain size. Consequently, the Atlantic interannual variability reflects the phase of the interannual North Atlantic SST tripole pattern rather than the traditional AMO on interdecadal time-scales, even though it is associated with the interannual AMOC and NAO-like pattern. As shown in Fig.3, the BKS SIC decline corresponds to a cold-warm-cold North Atlantic SST tripole (Fig. 3d) resembling the positive phase of AIV associated with an NAO⁺-like pattern (Fig. 3j). To some extent, the ENSO can significantly influence the interannual variability of the BKS SIC through modulating the phases of the AIV and NAO-like patterns. This is a new pathway proposed in this study.

To examine whether the La Niña, cold-warm-cold North Atlantic SST tripole and NAO⁺-like anomaly are optimal patterns that lead to the winter BKS SIC decline, here we show the first singular value decomposition (SVD) mode of the DJF-mean SIC anomaly over BKS associated with DJF-mean Z500 anomaly and the SVD time series of the BKS SIC during 1950-2019 in Figs. B3a-b. The dominant patterns of the co-variability of DJF-mean (c) Z500 (unit: gpm) and (d) SST (unit: K) anomalies with the BKS SIC decline obtained from the maximum covariance analysis are shown in Figs. B3c-d. It is found that the winter NAO⁺-like pattern with Ural blocking, La Niña and North Atlantic cold-warm-cold SST tripole are the optimal patterns for the winter BKS SIC decline. The results are consistent with our regression field results, which are shown in **supplementary Fig.4**. In the revised manuscript (**lines 247-250**), we have pointed out this point.

Figure B2. the EOF1 and PC1 time series of 10-yr high-pass filtered DJF-mean North Atlantic (20°-70°N, 80°W-0°) SST anomaly and 9-yr high-pass filtered DJF-mean North Atlantic (10°-70°N, 80°W-0°)

Figure B3. (a) Spatial pattern of the first singular value decomposition (SVD) mode of the DJF-mean SIC anomaly (color shading, unit: %) over BKS (30°-90°E, 65°-85°N) associated with DJF-mean Z500 anomaly and (b) the SVD time series of the BKS SIC during 1950-2019. (c, d) Dominant patterns of the co-variability of DJF-mean (c) Z500 (unit: gpm) and (d) SST (unit: K) anomalies with the BKS SIC decline obtained from the maximum covariance analysis (or SVD).

Question (4)

Moisture transport by the atmosphere and heat transport by the ocean are mentioned at many points (including the grand summary figure 5), but seemingly in passing without proper definition or explanation. It is unclear why moisture transport is more important than sensible heat transport vs. changes in the local energy budget (implied by the analysis of downward IR),

or whether thermodynamic impacts are more important than simple advection of sea ice.

Response:

Thank you for pointing out the need for clarification and explanation here. Luo et al. (2017, ERL) calculated the vertically-integrated horizontal water vapor fluxes under the different phases of NAO concurrent with Ural blocking. They found that the intrusion of moisture from the midlatitude North Atlantic to Barents-Kara Seas (BKS) through the Norwegian Sea is favored (suppressed) by Ural blocking concurrent with the positive (negative) NAO. In their work Luo et al. also found that the downward IR associated with the water vapor over the BKS due to the moisture intrusion plays a major role compared to the role of sensible and latent heat fluxes (SHF and LHF). Their results are shown in Fig. B4. One sees in this Figure that the values of the SHF and LHF (Figs. B4b-c) are smaller than that of the downward IR (Fig. B4a), thus suggesting that downward IR plays a major role compared to the roles of SHF and LHF. (This finding is consistent with the analysis of Lee et al. (2017)). The intensified downward IR (Fig. B4a) is related to increased water vapor over the BKS (Fig. B4d).

Lee, S., Gong, T., Feldstein, S.B., Screen, J. and Simmonds, I., 2017: Revisiting the cause of the 1989-2009 Arctic surface warming using the surface energy budget: Downward infrared radiation dominates the surface fluxes. *Geophys. Res. Lett.*, **44**, 10,654–10,661, doi: 10.1002/2017GL075375.

In our previous study (Chen et al., 2018, JC), we have discussed the role of horizontal winds associated with Ural blocking in the wind-induced sea ice drift out of the BKS and found that the role of the wind advection of sea ice is weak compared to the role of downward IR. In **lines 284-290** of this revised manuscript, we have cited our previous studies to support our results that downward IR plays a major role in the BKS SIC decline compared to the roles of SHF, LHF and sea ice advection.

The heat transport by the ocean is mainly represented by the heat transport anomaly like (Lien et al. 2017)

$$Q' = \rho c_p (\bar{V}T' + V'\bar{T} + V'T')$$

Where \bar{V} and \bar{T} are the mean volume transport and temperature, and the subscript “prime” denotes the perturbation.

Lien et al. (2017) found that enhanced heat transport entering the BKS is related to the positive phase of NAO and shows a notable interannual variability during 1990-2014. Such a heat transport

is an important factor influencing the BKS SIC decline, which is often referred to as the Atlantic warm water transport” (Arthun et al. 2012, 2019).

In lines 151-153 of our revised manuscript, we have changed “Atlantic warm water transport” into “Atlantic ocean heat transport” and given the definition of “moisture intrusion” and indicated that the BKS SIC variability is related to the Atlantic ocean heat transport and atmospheric moisture intrusion.

Figure B4. Time series of composite daily (a) downward IR, (b) sensible heat flux (SHF), (c) latent heat flux (LHF) and (d) total column water vapor (TCWV) anomalies averaged over the BKS and (e, f) composite daily vertically integrated moisture flux (MFC) (Taken from Luo et al. 2017).

Response to Presentation

Question (1):

The language is imprecise. It is my understanding that AIV is meant to refer to the first PCA of 9-year high pass Atlantic SSTs (line 111). These two terms are interchanged haphazardly through the manuscript, and as I'll show below, I'm still not sure what they mean. AIV introduced before it is defined, both in the abstract at line 43, and then 89-90, where it is associated with the AMOC, then line 107, and at last finally defined at line 111. Things are further complicated by efforts to take the first PCA of the total Atlantic SST record (which the authors call the AMO), and then high pass filter it (lines 125-126). I understand this produces a similar index, but it's not the same one. Frankly, I still don't understand what AIV is, as at line 118, the Atlantic SST PC1 time series is reported to be correlated at -0.25 with the Barents-Kara sea index, while at line 126, AIV is reported to be 0.38 correlated with Barents-Kara sea index. So are these indices not the same?

Response:

As indicated above, we have undertaken a major revision, rewritten the manuscript and have addressed the comment in connection precision of language. Of relevance to the point made above, we have pointed out that the first PCA of the 9-yr high-pass Atlantic SST reflects the interannual variability of the North Atlantic SST tripole anomaly, which is referred to Atlantic interannual variability (AIV). One reason for defining this parameter is to distinguish the traditional North Atlantic SST tripole because the traditional SST tripole includes the decadal components. The AIV is also different from the AMO with a period of 60-80 years, although the AMO is mainly linked to the interdecadal AMOC. The important point here is that in our paper the AIV reflects the interannual variability of the North Atlantic SST tripole anomaly and describes the different phases of the interannual North Atlantic SST tripole, even though it is associated with the interannual AMOC.

In our revision, we have simplified our previous descriptions about the 9-yr high-pass filter of the AMO index. Instead, we directly use the PC1 of the first EOF mode of the 9-yr high-pass filtered North Atlantic SST anomaly to define the AIV. In our previous manuscript, the difference of the correlation (-0.25 and -0.38) between the AIV and BKS SIC is due to the different definition of AIV. In the present revised manuscript, we have removed the discussion about the different definitions of AIV using the 9-yr high-pass filtered time series of the AMO index.

Question (2):

Long paragraphs make it hard to follow. Consider lines 130-160 which begins by abruptly introducing Ural mountain blocking, wanders through ENSO, and ends with analysis of NAO

anomalies. Keeping ideas compartmentalized in paragraphs would help the reader.

Response:

Thanks, we appreciate the point being made here. Transparency has now been enhanced by addressing this point.

Question (3):

The paper introduces needless complexity through a series of acronyms, many non-standard. Articles for Nature Communications should be written for a broad scientific audience. As an atmospheric scientist, I am familiar with the NAO, ENSO, and SIC, but was already struggling in the first paragraph where I had to digest BKS and AIV. We are soon barraged with AA (which is often associated with Alcoholics Anonymous) and AMO. Further along, STD (for standard deviation) is introduced but used a total of 4 times (twice to define it); this acronym also carries a less pleasant meaning.

Response:

Thanks for your suggestions. 'AA' and 'BKS' are very common acronyms in the polar scientific literature, but we do take your point. In the revised paper, we have removed the acronyms "AA", "STDs" and "AMO". Thus, in this revised manuscript we retained the abbreviations "AIV", "BKS" and the other acronyms.

Question (4):

Vague language is used at many points, e.g., at line 68 "Atlantic heat or warm water" , or line 107 "linked to ENSO probably through the AIV", as are adjectives like "mainly" etc., which lack quantitative rigor.

Response:

In the revision we have made major changes and present further quantitative estimates of the changes in the interannual variability of BKS SIC, AIV and ENSO from 1950-1989 to 1990-2019. We also removed some uses of "likely" or "mainly", because some quantitative evaluations have been given in previous studies (Luo et al. 2017; Lien et al. 2017).

Question (5):

I appreciate that English is not the native language of the lead authors of the study, and that the English use of definite vs. indefinite articles is particularly complicated. An English editor, or help from co-authors more fluent in the language, would help the clarity of the manuscript.

Response:

In this revised manuscript, we have improved the English expression in the revised manuscript.

Response to Reviewer #3's comments for the manuscript (NCOMMS-22- 22227)

We appreciate that the care and attention with which Reviewer #3 has assessed our submission. The very helpful comments have led us to undertake a major revision of the manuscript. The point-to-point responses to the comments are given below:

Question:

The authors investigate the causes of the interannual variability in Sea-Ice Concentration (SIC) over the Barents-Kara Seas (BKS) by implementing a 9-year high-pass filter to various fields with view to isolate the signal induced by ENSO variability. Although that the study is based solely on the analysis of statistical relationships and does not include any aspects of a modelling approach, I reckon that the work makes a very useful contribution to the investigation of the remote ENSO influence via the Atlantic atmospheric pathway. My main concern is that at many places the writing style is awkward to the extent that affects the discussion and significantly hinders the interpretation of some key results. This is particularly obvious in the section discussing the role of Ural blocking in tandem with the AIV. As a result, the manuscript needs to undergo extensive improvements in its writing style and presentation of results before it is considered for publication by Nature Communications. Also, the scientific merit of the work will be augmented in the case that the authors discuss the limitations of their work within the framework of other possible tropospheric or stratospheric pathways. Below, I indicate the parts of the manuscript where the discussion needs to be improved.

Response:

Thanks for providing useful comments on the improvement of this manuscript. We very much appreciate your view that the work makes a very useful contribution to the investigation of the remote ENSO influence via the Atlantic pathway'. In our revision we have devoted especial attention to improving the writing style and presentation of the results. We feel we have succeeded in these, and that now our messages are much more transparent. In addition, in the revised manuscript we found that the ENSO-BKS SIC connection is significantly strengthened from 1950-1989 to 1990-2019. We further attributed the enhanced interannual variability of the winter BKS SIC during 1990-2019 to increased variability of the ENSO amplitude in a warm climate. This is a surprise finding.

Responses to Major Comments:

Question (1):

L101-104 & Fig. 1b: The expression “interannual decline of the BKS SIC” is confusing and hinders the interpretation of Fig. 1b. Please rephrase so as to better convey the message that (on interannual time scales) the SIC over the BKS depletes during a La Niña-like SST anomaly over Pacific basin and a cold-warm-cold SST tripole pattern in the North Atlantic mid-high latitudes.

Response:

Thanks for this point. In the revised manuscript (lines 179-181), we have replaced “interannual decline of the BKS SIC...” by “the BKS SIC decreases during a La Niña-like SST anomaly over Pacific basin and a cold-warm-cold SST tripole pattern in the North Atlantic mid-high latitudes”.

Question (2):

L122-129: Here the authors present similar results but this time using unfiltered data. The discussion here should be a bit more detailed and explain the difference to what has been done so far and how the results has changed.

Response:

In the revised manuscript, we have removed the discussion about the 9-yr high pass filtered result of the AMO index taken from the website to avoid possible confusion.

Question (3):

Section “Role of Ural blocking regulated by Atlantic interannual variability in the BKS sea ice change.”

This section should be significantly revamped, there are too many instances where the writing style is loose and the discussion of results needs to be improved.

Response:

Thank you for this advice. This section has now been rewritten and undergone major revision.

Question “i) L169: Do the authors imply that “Composite analysis shows ...?”

Response:

We appreciate that our expression may have confused. The relevant text now reads “Composite analysis shows ...”. Some additional detailed description is now presented.

Question “ii) L169: “UB mainly occurs ...”. This expression gives a sense of frequency in the

occurrence of UB with respect to NAO or AIV. However, in Fig. 3ab, only time-mean composites of Z500 & SAT are presented. Do you imply that the UB anomaly is stronger during the positive phase of AIV? Please rephrase and clarify.”

Response:

In our revised manuscript, we have moved the results concerning the frequency of Ural blocking events concurrent with the different phases of NAO for the different phases of AIV shown in supplementary Figures to Fig. 5. The time-mean composite daily Z500 & SAT anomalies show that the composite UB is related to NAO⁺ (NAO⁻) during the positive (negative) phase of AIV, indicating that the positive (negative) phase of AIV favors UB-NAO⁺ (UB-NAO⁻) events. It does not mean that the UB anomaly is stronger during the positive phase of AIV than during the negative phase of AIV.

Question “iii) L169-170: “... the positive (negative) NAO phase or NAO+ (NAO-) during the positive (negative) phase of AIV...”. The sentence is confusing, it needs to be simplified and the use of too many parentheses should be avoided. What I do not understand is that while in Fig. 3ad, the time-mean composites of Z500 & SAT for the positive and negative AIV phase are presented, in the above sentence the expression “during the positive (negative) phase of AIV”. This adds another layer of classification with respect to NAO and AIV, which is not what is shown in Fig. 3 and described in the caption.”

Response:

Thank you for alerting us to the need for greater clarity here; the relevant parts of the text have now been rewritten. In the revised manuscript, we have defined the interannual NAO pattern associated with AIV or ENSO and the winter BKS SIC on interannual timescales as the winter NAO-like pattern, whereas the individual NAO events are sub-seasonal. Thus, UB-NAO⁺ or UB-NAO⁻ or UB-NAO^o events are defined as the combined events of sub-seasonal UB with the sub-seasonal NAO⁺ or NAO⁻ or NAO^o, even though most of Ural blocking events result from the decay of the sub-seasonal NAO⁺ events due to energy dispersion. In this revised manuscript, the results concerning the frequency of UB-NAO⁺, UB-NAO⁻ and UB-NAO^o events are presented based on the different phases of the AIV.

Question “iv) L175-177: “We also note that the UB-related Eurasian cold anomaly shows a more westward shift for the positive phase of AIV than for its negative phase”. I do not see and westward shift but only an eastward shift during AIV+ in Fig. 3b.”

Response:

Thank you for picking up this error. As you said, this should have referred to an eastward shift. Please note, however, that in the revised manuscript we have removed this discussion, even though an eastward shift can be seen for the positive phase of AIV.

Question “v) L181-183: “But during the negative phase winter of the AIV the UB-NAO+ events have the almost same magnitude frequency...”. Do you refer again to the frequency of UB or the intensity/amplitude of the UB anomalies? “

Response:

In this revised manuscript we have made this clearer. We have re-calculated the frequencies of UB-NAO⁺, UB-NAO⁻ and UB-NAO^o events based on the different phases of AIV and shown their results in Fig.5. It is found that UB-NAO⁻ (UB-NAO⁺) events are favored (suppressed) by the negative phase of AIV. Their difference in intensity/amplitude between the positive and negative phases of AIV can also be found in Figs. 6a, c.

Question“vi) L185-186: Given the above-mentioned problems the results concerning the classification of UB occurrence or intensity with respect to NAO and AIV should be carefully rephrased so as they are assessed during a possible second revision.”

Response:

We have rephrased our classification in our revision. The UB events are classified according to the different phases of NAO events on sub-seasonal timescale (10-20 days). Such a classification depends on the different phases of AIV on interannual timescales. Thus, we show UB-NAO⁺, UB-NAO⁻ and UB-NAO^o events in Fig. 5 for the different phases of AIV to show that UB-NAO⁺ (UB-NAO⁻) events are more frequent during the positive phase of AIV than during the negative phase of AIV.

Question (4):

L204-208 & Fig. 4. The results presented here are interesting but again the discussion should be improved.

Response:

Thanks for the encouraging comment. We have now made clearer the linkage of the Atlantic Hadley cell and Walker cell to ENSO in **lines 322-330**.

Question “i) Perhaps I miss something, but how are the fields of divergent horizontal and vertical velocity calculated? Do they correspond to some kind of normalized anomalies with respect to

climate? Please clarify in the text and in the caption”.

Response:

In this revised manuscript, our calculation is based on the anomaly fields of divergent horizontal and vertical velocity relative to climatological mean. This has been described in the text and in the caption.

Question “ii)L203-212: Please avoid the excessive use of parentheses in the text that hinders the interpretation of results.”

Response:

In this revised manuscript, we have made a revision to avoid the excessive use of parentheses.

Question “iii) L214-217: This sentence is confusing (also too many parentheses where the p-values are explained). Again, unfiltered data is introduced but the methodology should be explained in a more careful and detailed manner.”

Response:

In this revised manuscript, we have removed the discussion of the unfiltered data. Moreover, the explanation about the p-values has been added in Methods.

Question (5):

L246-250: Summary

In this work the remote impact of ENSO on the interannual variability of SIC over the BKS region via the Atlantic path is presented. I particularly like the schematic in Fig. 5. However, the authors should enhance the discussion by making comments concerning the remote impact of ENSO via the stratospheric pathway or even through influencing the mid-latitude wave-guide over the Pacific that could be later communicated over the North Atlantic basin. In other words, the work would be more complete in the case that the authors discuss in more details the limitations of their work within the framework of other possible tropospheric or stratospheric pathways.

Response:

In the revised manuscript (lines 401-406), we have strengthened our discussion about the schematic diagram and pointed out the limitations of our framework particularly in connection to tropospheric or stratospheric pathways.

Responses to Minor comments

Question:

- 1) L51: “on trend” is a bit awkward. Please rephrase.
- 2) L52: “main driver”
- 3) L55: “over the Northern Hemisphere continents”?
- 4) L55: The expression “decline trend” is confusing. Please rephrase.
- 5) L122: Again, the loose use of “decline” is confusing. Perhaps it should be replaced by “We also note that the BKS SIC depletion (on interannual timescales) occurs in tandem with a combination...”.
- 6) L242: “...remote modulation of ENSO in the Pacific basin...” Do you mean remote modulation of the SIC anomalies by ENSO? If yes, please correct throughout the manuscript.

Response:

Thanks for alerting us to these various ambiguities. All have now been addressed.

Peer Review Comments, second round review –

Reviewer #1 (Remarks to the Author):

In the revised manuscript, the authors present a new finding that the increased interannual variability of winter BKS SIC since 1990 is likely related to the enhanced variability of the ENSO amplitude in a warm climate. The closer connection (higher correlation coefficient) between ENSO and BK SIC in a warm climate in recent decades helps to analyze the associated physical mechanisms and improve our understanding of the predictability of the BKS SIC. I am generally satisfied with the progress made by the author and the response to me.

Here, I have a few small suggestions.

1. Line 328: Here, "the region" is a bit vague for me, do you mean the weakened Walker circulation in the equatorial Atlantic? For LN, the mean Walker circulation is enhanced.

Question 7: Here, I mean that after removing the linear impacts of ENSO on AIV, instead of linear trends, the residual AIV variability significantly attenuates, especially for 1990-2019. It may further confirm the importance of ENSO for AIV or BKS SIC. However, the authors have amended and used a reasonable expression.

Reviewer #3 (Remarks to the Author):

Second review on the submitted article "Origins of the interannual variability of sea ice over the Barents-Kara Seas: Atlantic pathway of the ENSO modulation" by B. Luo et al

The authors have adequately treated most of the issues raised in my previous assessment during the previous round of revisions. Therefore, I recommend that the manuscript is accepted for publication by Nature Communications in its present form.

Response to Reviewer #1's comments for the manuscript (NCOMMS-22-22227)

The authors would like to thank Reviewer #1 for his/her helpful comments. We have made a minor revision according to your comments. The point-to-point responses to Reviewer #1's comments are given below:

Response to general statement:

Question:

In the revised manuscript, the authors present a new finding that the increased interannual variability of winter BKS SIC since 1990 is likely related to the enhanced variability of the ENSO amplitude in a warm climate. The closer connection (higher correlation coefficient) between ENSO and BK SIC in a warm climate in recent decades helps to analyze the associated physical mechanisms and improve our understanding of the predictability of the BKS SIC. I am generally satisfied with the progress made by the author and the response to me.

Response:

We appreciate your useful comments in improving the quality of the manuscript. We have made a minor revision based on your further comments.

Response to Minor comments:

Question:

1. *Line 328: Here, "the region" is a bit vague for me, do you mean the weakened Walker circulation in the equatorial Atlantic? For LN, the mean Walker circulation is enhanced.*

Response:

In **line 330** of the revised manuscript, we have revised the vague writing and changed "the region" into "the equatorial Atlantic". For La Niña, the mean Walker circulation is weakened. In contrast, for El Niño, the mean Walker circulation is enhanced.

Question:

2. *Question 7: Here, I mean that after removing the linear impacts of ENSO on AIV, instead of linear trends, the residual AIV variability significantly attenuates, especially for 1990-2019. It may further confirm the importance of ENSO for AIV or BKS SIC. However, the authors have*

amended and used a reasonable expression.

Response:

In the revised manuscript, the correlation between 9-year high-passed ENSO and AIV reflects the linear impacts of ENSO on AIV or the importance of ENSO for AIV or BKS SIC. Their correlation coefficient increases -0.049 ($p>0.1$) during 1950-1989 to -0.51 ($p<0.01$) during 1990-2019. In this case, the residual AIV variability significantly attenuates, especially for 1990-2019, after removing the linear impacts of ENSO on AIV. Thus, in **lines 131-136** of our revised manuscript, the description about the relationship between ENSO and AIV is reasonable.